# Parabolic avalanche scaling in the synchronization of cortical cell assemblies

Elliott Capek [1,4], Tiago L. Ribeiro [1,4], Patrick Kells[1], Keshav Srinivasan[1,2], Stephanie R. Miller [1], Elias Geist[1], Mitchell Victor[1], Ali Vakili[1], Sinisa Pajevic [1], Dante R. Chialvo [3] & Dietmar Plenz [1] ✉

Neurons in the cerebral cortex fire coincident action potentials during ongoing activity and in response to sensory inputs. These synchronized cell assemblies are fundamental to cortex function, yet basic dynamical aspects of their size and duration are largely unknown. Using 2-photon imaging of neurons in the superficial cortex of awake mice, we show that synchronized cell assemblies organize as scale-invariant avalanches that quadratically grow with duration. The quadratic avalanche scaling was only found for correlated neurons, required temporal coarse-graining to compensate for spatial sub-sampling of the imaged cortex, and suggested cortical dynamics to be critical as demonstrated in simulations of balanced E/I-networks. The corresponding time course of an inverted parabola with exponent of $\chi = 2$ described cortical avalanches of coincident firing for up to 5 s duration over an area of 1 mm$^2$. These parabolic avalanches maximized temporal complexity in the ongoing activity of prefrontal and somatosensory cortex and in visual responses of primary visual cortex. Our results identify a scale-invariant temporal order in the synchronization of highly diverse cortical cell assemblies in the form of parabolic avalanches.

Neuronal synchronization is fundamental to many theories of the cerebral cortex. Cortical neurons, by preferentially integrating recurrent activity from neighboring cells[1] as well as from distant inputs, support at least two main synchronization dynamics: oscillations[2] and cascades, the latter in the form of waves[3,4], synfire chains[5,6], and neuronal avalanches[7,8]. Cortical cascades, in which a neuronal group that fires coincident spikes facilitates synchronization in downstream neurons through select, converging connections, suggest a particular powerful mechanism to establish robust, yet flexible information processing in the cortex[5–7]. However, both, the growth of cascades and their overall temporal profile, i.e., the number of neurons or spikes encountered over time, have been found in simulations to be variable, challenging their ability to reliably transmit information within the cortical network[5,7,9–11]. This problem has been particularly prominent for neuronal avalanches[7], which represent highly diverse, scale-invariant cascades of neuronal activity predominantly found in the superficial layers of cortex. Avalanches are readily observed in the local field potential[12], selectively engage single neurons[13,14], and carry high information capacity[15,16]. Yet, it is currently not clear whether avalanches, when measured at the single cell level in vivo[8,17–19], do exhibit robust neuronal synchronization that unfolds in a predictive manner.

Traditionally, the hallmark of neuronal avalanches has been their scale-invariant spatiotemporal statistics quantified by power laws in size and duration[7] with exponents $\alpha \approx 3/2$ and $\beta \approx 2$, respectively. More recently, their temporal profile, i.e., how the size of an avalanche unfolds in time, has been suggested to discriminate between models of avalanche generation. Specifically, the scaling of their mean size with duration and their universal, duration-invariant temporal profile[20–22], can both be captured in a single scaling exponent $\chi$. For models of avalanches lacking interactions, generated by noise, or those found

[1]Section on Critical Brain Dynamics, National Institute of Mental Health, Bethesda, MD, USA. [2]Department of Physics, University of Maryland, College Park, MD, USA. [3]CEMSC3, Escuela de Ciencia y Tecnologia, UNSAM, San Martín, P. Buenos Aires, Argentina. [4]These authors contributed equally: Elliott Capek, Tiago L. Ribeiro. ✉e-mail: plenzd@mail.nih.gov

near a 1st-order phase-transition, $\chi$ ranges between 1 and 1.5 and temporal profiles are non-parabolic, e.g., flat, semi-circle, or even sawtooth like[23,24]. In contrast, for avalanches that unfold according to a critical branching process, a close approximation for synchronized cascades[7,25] $\chi = 2$ and avalanche profiles are parabolic[11,20–22], similar to what can be found close to 2nd-order phase transitions that fall into the directed percolation universality class[26] (see also ref. 27). This relationship has recently been shown for LFP-based avalanches in non-human primates[28], which suggests that avalanches describe rapid, scale-invariant unfolding of neuronal synchronization, i.e., coincident spiking. However, when measured at the cellular level in the mammalian cortex, neuronal avalanches exhibited $\chi$ between ~1 and 1.3 with non-parabolic, even asymmetrical profiles[17,19,29,30], disenfranchising neuronal avalanches as a potential framework for cortical synchronization.

Contrary to those reports, we demonstrate here, at cellular resolution using 2-photon imaging (2PI) in the cortex of awake transgenic mice, that $\chi = 2$ for neuronal avalanches which identifies a scale-invariant, temporal profile in the form of a symmetrical, inverted parabola that maximizes temporal complexity and increases temporal correlation in cortical population activity. Our findings establish a robust scaling relationship for the synchronization of cortical cell assemblies in the form of parabolic avalanches.

## Results

### Synchronized assemblies in prefrontal cortex exhibit quadratic avalanche scaling in mean size vs. duration

We studied neuronal synchronization in the prefrontal cortex of awake mice during resting and spontaneous locomotion using 2PI (Fig. 1). Chronic implantation of a prism[31] allowed us to simultaneously image from ~200 to 300 neurons within a window of 450 × 450 μm across the fissure in the contralateral, intact superficial layers of anterior cingulate and medial prefrontal cortex (ACC, mPFC) at a temporal resolution $\Delta t = 22$ ms (~45.5 Hz frame rate; Fig. 1a; Supplementary Fig. 1).

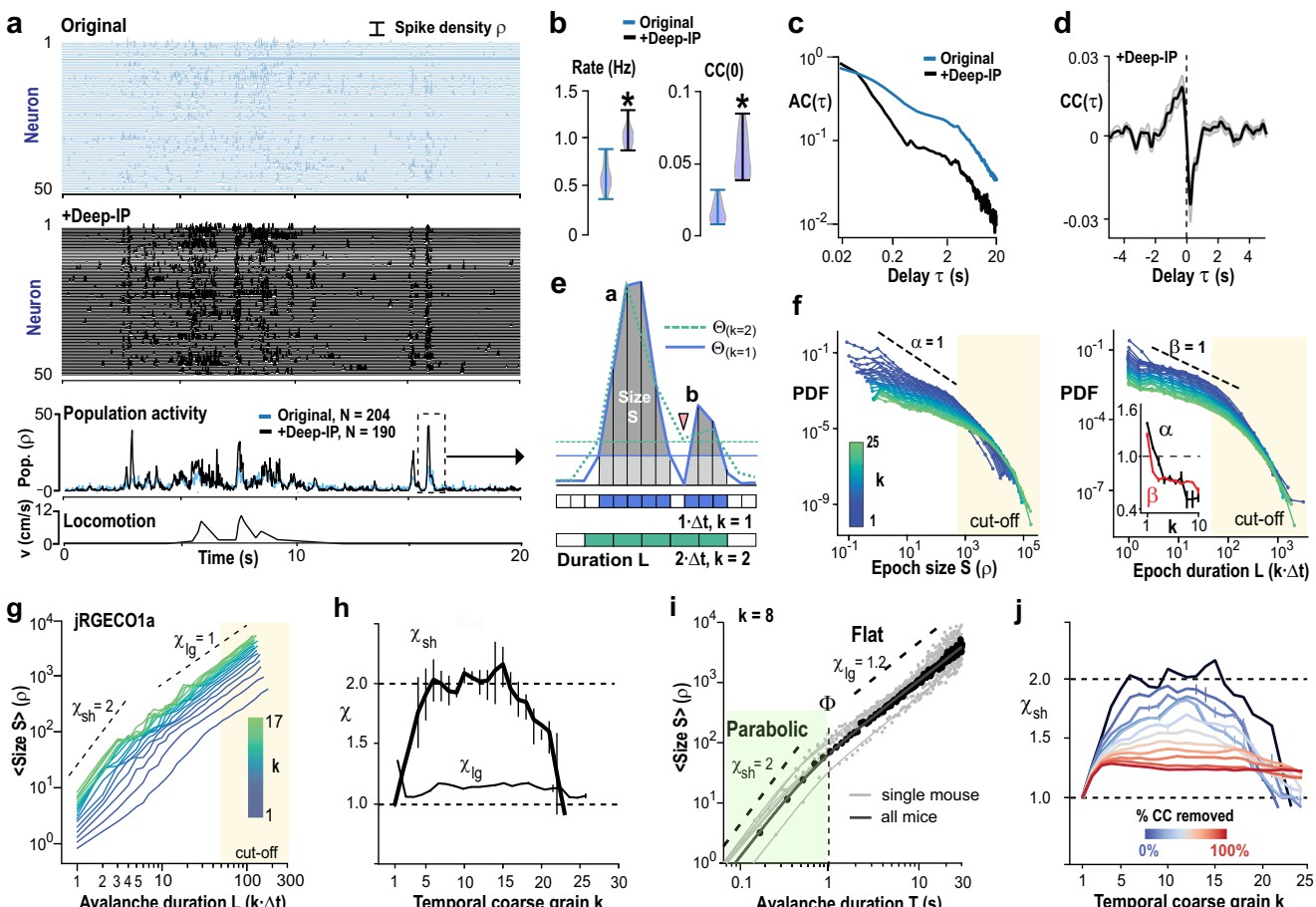

**Fig. 1 | Neuronal synchronization exhibits avalanche scaling of $\chi = 2$ in prefrontal cortex. a** Ongoing neuronal activity in mPFC/ACC of awake mice exhibits epochs of increased population activity (columnar organization; *original*) enhanced after deep-interpolation (+*Deep-IP;*2PI, jRGECO1a). Spike density raster of 50 randomly selected neurons each (single mouse). Middle: Full population activity (*Pop*). Bottom: Locomotion velocity (*v*). **b, c** Deep-IP increases neuronal firing (*Rate*), cross-correlation (*CC*), and sharpens population activity transients visible in a faster decay of the autocorrelation (*AC*). **d** Locomotion onset is preceded by population activity increases (*CC*). **b–d** Corresponding experiment in (**a**). **e** Sketch of continuous epoch in population synchrony per $\Delta t$ above 'hard' threshold $\Theta(k = 1)$ (blue line). Corresponding size $S$ (gray) and duration $L$ in multiples of time bins (blue). A 'soft' threshold discards the subthreshold area (light gray; $k = 1$). Note that the two suprathreshold epochs $a$ and $b$, separated at $1 \cdot \Delta t$ (arrowhead), will merge upon temporal coarse-graining ($2 \cdot \Delta t$, $k = 2$; green dotted line), despite an increase in synchrony requirement per $\Delta t$ ($\Theta(k = 2)$; broken green line). **f** Suprathreshold epochs fulfill the criteria of neuronal avalanches. Epoch size $S$ (left) and duration $L$ (right) exhibit power laws robust to temporal coarse-graining ($k = 1$–25; color code). Note cut-offs for $S > 1000$ and $L > 50$ (beige areas). *Inset*: Corresponding decreasing slopes $\alpha(k)$ and $\beta(k)$ cross the value of 1. **g** Change in mean size <$S$> vs. duration $L$ for $k = 1$–17. Slope estimates for $L = 1$–4 ($\chi_{sh}$) and $L \geq 10$ ($\chi_{lg}$). **h** Summary of $\chi_{sh}$ and $\chi_{lg}$ as a function of $k$. SD from $n = 10$ bootstrapping repeats. **i** For 'parabolic' avalanches with durations of $T < 1$ s, $\chi_{sh} = 2$ at $k = 8$, whereas 'flat' avalanches exhibit $\chi_{lg} \cong 1.3$ and $T > 2$ s. Avalanches combined for *single mice* and for *all* mice. Solid line: fit. **j** $\chi_{sh} = 2$ reflects inter-neuronal correlations and $\chi_{sh} \ll 2$ when neurons are made independent by random temporal shifts (*black*; see (**h**); 10 surrogates/condition). **f–i** Data pooled from $n = 5$ mice and 17 experiments; jRGECO1a. Values presented as mean or mean ± SD.

Machine-learning based deep-interpolation[32] (Deep-IP) markedly improved the extraction of spike densities from each neuron yielding significantly higher average firing rate and pairwise correlation among neurons ($n = 5$ mice; Fig. 1b; Supplementary Fig. 2). We summed activity across all neurons to obtain the time course of population activity. This time course exhibited rapid transients during which neurons showed coincident firing, i.e., spike synchronization in their activities, either spontaneously or prior to self-initiated locomotion (Fig. 1a, c, d).

We examined these epochs of synchronization systematically (1) by requiring the summed spike density within a time of $\Delta t$, i.e., population activity, to be larger than a threshold $\Theta$, and (2) by concatenating successive suprathreshold population events (Fig. 1e). This commonly employed approach (e.g., ref. 33) to identify contiguous periods of significant network activity is sensitive to spatial subsampling and noise. Spatial subsampling prematurely terminates contiguous periods by missing neuronal activity, which we compensated for by systematically relaxing our concatenation criterion. Noise, on the other hand, reduces the ability to correctly identify suprathreshold periods. This error can be reduced by increasing $\Theta$, which selects higher coincident neuronal activity in the network, i.e., higher synchronization. Accordingly, suprathreshold population events, identified at $\Delta t$, were concatenated at the temporally coarse-grained resolution of $k \cdot \Delta t$, $k = 1, ..., k_{max}$, $k_{max} < 40$. We further set $\Theta(k) = -2SD(k)$ of the z-scored distribution in number of epochs, which increases the absolute value of $\Theta$ with $k$ (Fig. 1e; Supplementary Fig. 3a, b; see "Methods"). We found that, independent of temporal coarse-graining, epochs in population synchrony exhibited the hallmark of neuronal avalanches[7,8,12,14] with their sizes, $S$, i.e., summed suprathreshold spike densities (see "Methods" and Supplementary Fig. 6), and durations, $L$, i.e., the number of time bins or generations per epoch, distributed according to power laws (Fig. 1f; Supplementary Figs. 3c, 4). The high diversity in size and duration of avalanches was quantified by the corresponding slopes $\alpha$ and $\beta$ being smaller than 2 and rapidly crossing the value of 1 upon temporal coarse-graining, demonstrating a broadening of the corresponding distributions despite robust size and duration cut-offs of $S > 10^3$ and $L > 50$, respectively (Fig. 1f; inset).

Next, we derived the scaling exponent, $\chi$, which describes the power-law dependence between avalanche size and duration, visualized by plotting the mean avalanche size for avalanches of a given duration[20–22]. We found that $\chi$ for short-lasting, i.e., few-generation ($L < 5$) avalanches, $\chi_{sh}$, significantly increased with temporal coarse-graining, reaching maximal values around 2 (Fig. 1g, h). In contrast, $\chi$ for long-lasting, i.e., many-generation ($L > 10$–$30$) avalanches, $\chi_{lg}$, remained close to 1 independent of the temporal scale at which epochs were observed (Fig. 1g, h). The corresponding transition marks the scaling range, $\Phi$, obtained by fitting power law functions to these two regimes (see "Methods") and demonstrates that avalanches with absolute duration $T = L \cdot k \cdot \Delta t$ shorter than 1 s ($\Phi = 0.52 \pm 0.21$; $n = 5$) exhibited a quadratic, rapid growth in size as they unfolded in the network, which was not found for longer lasting avalanches ($T > \sim 2$ s; Fig. 1i). For simplicity, we define short-lasting avalanches with quadratic growth as 'parabolic' avalanches, in contrast to long-lasting, 'flat' avalanches.

Parabolic avalanches strongly depended on spatial correlations in the network. First, parabolic avalanches were abolished when neuronal time series were randomly shifted in time (Supplementary Fig. 2g) and secondly, $\chi_{sh}$ monotonically decreased to values lower than 2 with the percentage of spatial correlations removed (Fig. 1j) or percentage of uncorrelated spikes introduced (Supplementary Fig. 5a).

Our approach was robust to subsampling given that up to a 50% reduction in our neuronal sampling fraction, $f$, still allowed us to robustly recover $\chi_{sh} \cong 2$ due to a concomitant decrease in minimal requirement for coincident activity $\Theta$ (Supplementary Fig. 5b). The identification of $\chi_{sh} \cong 2$ was also robust to different forms of thresholding. Thresholding of an activity time series introduces a systematic error in the scaling estimate of $\chi$[34]. Our analytical derivation (Supplementary Notes) and corresponding data analysis demonstrates $\chi_{sh} \cong 2$ to be within the boundaries of both approaches (Supplementary Fig. 6a, b) and to be robust over a large range of thresholds (Supplementary Fig. 6c). Separating population activity into resting and locomotion periods did not change our finding of $\chi_{sh} \cong 2$ despite a significant increase in neuronal firing rate when mice were spontaneously running (Supplementary Fig. 7).

We repeated our findings using GCaMP7s, which has a slower decay time constant and less tissue penetration capability for 2PI compared to jRGECO1a[35] (Fig. 2a–d; $n = 3$ mice; $\Delta t = 22$ ms). To focus on population synchrony without applying Deep-IP, we removed weakly correlated neurons after z-scoring each neuron's correlation with the population activity (Supplementary Fig. 8a–d; 10–30% of neurons removed per recording; see "Methods"). In line with our findings using jRGECO1a, suprathreshold epochs in population activity exhibited the hallmark of avalanches (Supplementary Fig. 8e–g). Temporal coarse-graining revealed parabolic avalanches at $k \cong 16$ with duration of ~0.3–1.8 s that demonstrated $\chi_{sh} \cong 2$ (Fig. 2a–c), which was sensitive to contributions from uncorrelated cells or shuffling (Fig. 2d; Supplementary Fig. 8h). In contrast, flat avalanches exhibited $\chi_{lg} \cong 1$ independent of temporal coarse-graining (Fig. 2b).

Our findings suggest that the two categories of parabolic and flat avalanches might arise from experimental shortcomings in accurately tracking neuronal avalanches in the cortical network. Sizes and durations of flat avalanches overlap with the cut-off regimes in the corresponding duration and size distributions (Fig. 1g, Fig. 2a; Supplementary Fig. 8e, f). Such cut-offs reflect finite-size effects from the recording field of view (FoV), which greatly impacts avalanche measures[36]. Indeed, when employing mesoscope imaging in somatosensory cortex of Thy1 transgenic mice expressing GCaMP6s (Fig. 2e, f), the maximal duration of parabolic avalanches, i.e., $\Phi$, correspondingly increased to ~5 s with a FoV of ~1 mm$^2$ (Fig. 2g, h; Supplementary Fig. 9).

Our results establish that synchronization of ongoing cell assemblies in frontal and somatosensory cortex organizes as parabolic avalanches with rapid, quadratic expansion in coincident firing over time.

## Temporal coarse-graining recovers $\chi = 2$ for critical branching process under subsampling conditions

Next, we use simulations to explore the hypothesis that temporal coarse-graining can recover parabolic avalanche synchronization in critical networks that are incompletely observed, i.e., spatially subsampled. We chose a network of $N = 10^6$ binary, probabilistic, integrate, and fire neurons (80% excitatory, E; 20% inhibitory, I) and all-to-all connectivity (Fig. 3a) with excitatory and inhibitory connectivity matrices $W_{EE} = W_{IE} = J$ and $W_{II} = W_{EI} = -g \cdot J$ that were constant. We set $g$ to 3.5 and obtained an E/I-balance that supported avalanche dynamics in the fully sampled system with critical exponents $\alpha = 3/2$, $\beta = 2$, and $\chi = 2$ approximating a critical branching process triggered by external Poisson inputs[37] (Supplementary Fig. 10). We found that temporal coarse-graining recovered $\chi_{sh} = 2$ for few-generation avalanches over a wide range of subsampling conditions denoted by the neuronal sampling fraction, $f = [0.01\%, ..., 100\%]$ and population activity threshold, $\Theta = [0, ..., <20,000$ spikes per time unit]. We exemplify these findings in Fig. 3b–f using a sampling fraction of $f = 0.1\%$ and $\Theta = 1$, i.e., the minimum requirement of 2 spikes per time unit. In line with our experimental findings, temporal coarse-graining of subsampled avalanches reduced the power law slopes in size and duration and identified the mean size vs. duration power law relationship of $\chi_{sh} \cong 2$ for parabolic avalanches (Fig. 3e). We note that successive avalanches are uncorrelated in the model, which reduces $\chi_{sh}$ for very large $k$ independent of the total simulation time (Supplementary Fig. 11). As found

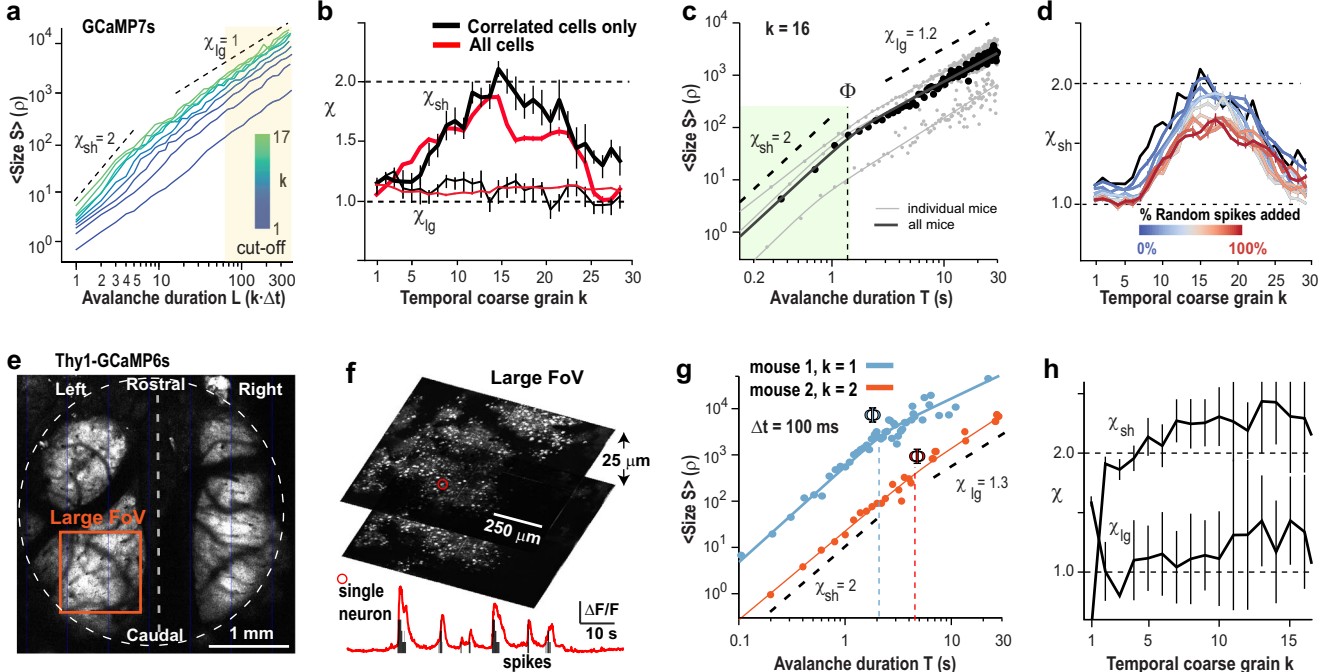

**Fig. 2 | Avalanche scaling of $\chi = 2$ for ongoing neuronal synchronization is robust to numerous experimental conditions. a–d** Ongoing activity in mPFC/ACC using GCaMP7s exhibits parabolic avalanches in the subset of synchronized neurons. **a** $\chi_{sh} \cong 2$ is identifiable in the subpopulation of synchronized neurons for $k = 16$ (data pooled from $n = 3$ mice over 27 recordings). **b** Corresponding value of $\chi_{sh}$ (thick line) and $\chi_{lg}$ (thin line) as a function of $k$ for all (red) and synchronized neurons only (black). SD from $n = 10$ bootstrapping repeats. **c** Corresponding scaling with $\chi_{sh} \cong 2$ for avalanches of duration -0.3–1.8 s ($\Phi$) at $k = 16$. **d** Addition of random spikes, which mimics the presence of noisy, non-synchronized neuronal activity degrades the capacity to recover $\chi_{sh} \cong 2$ (GCaMP7s). Black: see (**b**); mean ± SD over 10 surrogate sets per condition. **e–h** Ongoing activity in somatosensory cortex measured with transgenic mice intrinsically expressing GCaMP6s imaged using a mesoscope. **e** Overview of mesoscopic imaging of large brain area. Note expression of GECIs obscured by large blood vessels. Broken line: midline. *FoV*: field of view (orange square). **f** High-resolution image of the FoV in (**e**) and corresponding period of spontaneous fluctuations in relative fluorescence (red; au) and spikes (black) in a single neuron (red circle). **g** Scaling regime of $\chi_{sh}$ increases with FoV. For a large FoV, $\chi_{sh} \cong 2$ holds over for avalanches of duration 0.1–5 s ($\Phi$) at $k = 1$–2 ($n = 2$ mice from 2 recordings). **h** Corresponding average value of $\chi_{sh}$ and $\chi_{lg}$ as a function of $k$ (soft thresholding; mean ± SD). Black dashed lines: Guide to the eye.

for our 2PI data, noise reduces the recovered maximum value of $\chi_{sh} \leq 2$ without shift in $k$ (Supplementary Fig. 12a) and better tracking of spiking activity shifts the recovery of $\chi_{sh}$ towards smaller $k$ as found for deep-interpolation (Supplementary Fig. 12b). In general, we found that sampling of fewer neurons (low $f$) and less sensitivity (high $\Theta$) required more temporal coarse-graining to recover $\chi_{sh} \cong 2$ (Fig. 3e). The model also demonstrated that the scaling range $\Phi$ for which $\chi_{sh} \cong 2$ reflects a finite-size effect that is reliably recovered within the accuracy of the temporal coarse-grain (Fig. 3f). As expected, $\chi_{lg} \ll 2$ for many-generation, i.e., flat, avalanches, which, similarly to our data, are located in the cut-off of the corresponding size and duration distribution (Fig. 3c, d).

Importantly, our simulations show that the recovery of $\chi_{sh} \cong 2$ under spatial subsampling conditions is only possible when the network exhibits critical dynamics, whereas $\chi_{sh}$ remains $\cong 1$ for subcritical dynamics regardless of temporal coarse-graining (Fig. 3g; Supplementary Fig. 13). These simulations support the view that our 2PI data represent spatially subsampled activity of critical dynamics in cortex.

### Evoked synchronization in primary visual cortex exhibits $\chi = 2$ avalanche scaling

Evoked visual and auditory responses in primary visual (V1) and auditory (A1) cortex have been found to organize as neuronal avalanches exhibiting power laws in size and duration distribution[17,18]. Yet, the relationship between the duration of evoked avalanches and their mean size, $\chi$, has been reported to be between 1–1.3 when measured at typical 2PI frame rates of, e.g., 33 Hz[17,19]. We studied $\chi$ for neuronal responses in superficial layers of V1 to large-field gratings drifting in 8 directions using GCaMP7s and in conjunction with Deep-IP and temporal coarse-graining (Supplementary Fig. 14a–d; $n = 2$ mice and 3

recordings). In line with our findings for ongoing activity, temporal coarse-graining recovered $\chi_{sh} \cong 2$ for few-generation avalanches of durations $T = -0.1$–1.6 s (Fig. 4a). These parabolic avalanches were abolished by trial-shuffling demonstrating their dependence on spatial correlations within each trial (Supplementary Fig. 14e–g). We then extended our analysis to the publicly available Allen-Institute data set on V1 evoked responses in superficial layers of cortex of the awake mouse ($n = 8$ mice, 5 males/3 females, 8 recordings for drifting gratings and movies each). This data set used GCaMP6f, which preferentially reports action potential bursts[38] equivalent to applying a high, local $\Theta$ in our analysis. Responses to drifting gratings as well as to movies demonstrated $\chi_{sh} \cong 2$ under temporal coarse-graining for few-generation avalanches, which again was abolished after trial and temporal shuffling, respectively (Fig. 4b; Supplementary Fig. 14h). Our analysis confirms and extends $\chi \cong 2$ from ongoing activity to evoked parabolic avalanches during sensory processing.

### Avalanche scaling of $\chi = 2$ maximizes temporal complexity and identifies a scale-invariant inverted parabola in cortical synchronization

Next, we explored how the initial increase and subsequent decrease in $\chi_{sh}$ with temporal coarse-graining relates to the temporal organization of neuronal population activity. Accordingly, we quantified the temporal complexity in population activity by dividing the thresholded population activity time course into $D \in 4, \ldots, 7$ successive temporal bins of duration $k \cdot \Delta t$. For each pattern depth, $D$, and coarse-graining factor $k$, we calculated the pattern complexity, $Cx$, quantified as the number of different temporal sequences of length $D$ (ref. 39; see "Methods"). We found that $Cx$ peaked near the $k$ for which $\chi_{sh} = 2$, for

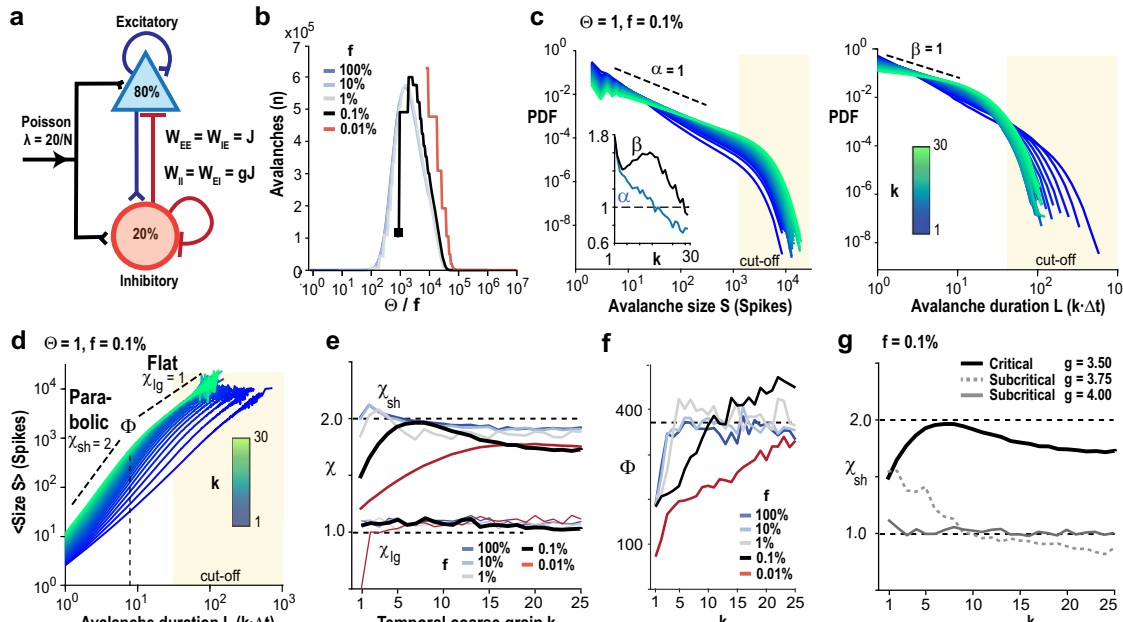

**Fig. 3 | Temporal coarse-graining recovers avalanche scaling of $\chi = 2$ under subsampling conditions when the network is critical. a** Sketch of the neuronal network with $N = 10^6$ neurons (80% excitatory, E; 20% inhibitory, I) and external Poisson drive of rate $\lambda = 20/N$ per time step. The E/I balance is controlled by the scalar $g$, which scales inhibitory weight matrices $W_{II} = W_{EI}$ as a function of excitatory weight matrices $W_{EE} = W_{IE} = J$. **b** Change in number of avalanches as a function of threshold $\Theta$ normalized by sampling fraction $f$. **c** Power law in size (left) and duration (right) distributions for avalanches become shallower with temporal coarse-graining $k$. Note cut-off regimes for $S > \sim 10^3$ and duration $L > \sim 50$. *Inset*: Corresponding slopes $\alpha(k)$ and $\beta(k)$. **d** Temporal coarse-graining uncovers $\chi_{sh} = 2$ for short-duration avalanches ($L = 1$–10), whereas $\chi_{lg}$ remains $-1$–1.2 for long-

duration avalanches ($L > 10$). **e** Summary of change in $\chi_{sh}$ and $\chi_{lg}$ with $k$ for 5 sampling fractions $f$. A decrease in $f$ requires a higher $k$ to recover $\chi_{sh} = 2$. Note failure of recovery for very low $f$. $\chi_{lg}$ does not depend on $k$. **f** Temporal coarse-graining recovers avalanches up to the finite-size cut-off of $\Phi \cong 400$ time steps. Note plateau in maximal scaling range $\Phi$ for $\chi_{sh} = 2$ plotted in simulation time steps as a function of $k$. **g** Temporal coarse-graining recovers $\chi_{sh} = 2$ for critical network dynamics but fails for subcritical dynamics. Note that weak subcritical ($g = 3.75$) and critical conditions ($g = 3.5$) can exhibit similar $\chi_{sh}$ at the original temporal resolution yet diverge with temporal coarse-graining. Broken, black lines: visual guide to the eye. Results obtained from $T = 10^8$ simulation time steps.

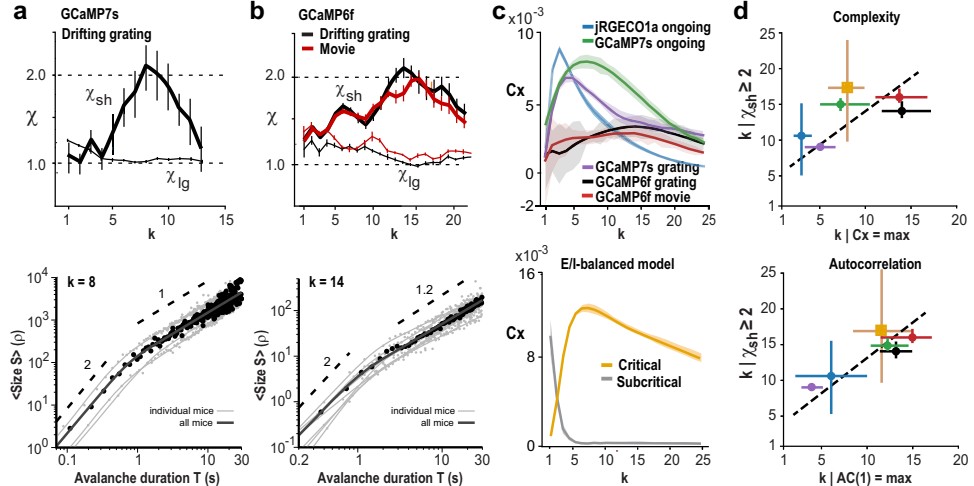

**Fig. 4 | Visually evoked neuronal assemblies in V1 exhibit avalanche scaling of $\chi = 2$ at which complexity and temporal correlations are maximized. a** Top: temporal coarse-graining uncovers $\chi_{sh} \cong 2$ for visually evoked, few-generation avalanches ($L = 1$–4). In contrast, many-generation avalanches ($L = 10$–20) exhibit $\chi_{lg} \sim 1$ at any temporal coarse-graining (Data pooled from $n = 2$ mice over 3 experiments; SD from $n = 10$ bootstrapping repeats; 45.5 Hz; GCaMP7s; +Deep-IP). Bottom: corresponding scaling of $\chi_{sh} = 2$ holds for evoked avalanches of duration $T < \sim 1$ s for $k = 11 \pm 8$ (mean ± SD over 3 independent experiments). **b** Top: recovery of $\chi_{sh} = 2$ in V1 during viewing of drifting gratings (black) or movies (red; 33 Hz, GCaMP6f; $n = 8$ mice over 8 experiments; SD from $n = 10$ bootstrapping repeats; Allen Institute data set). Thin lines: $\chi_{lg}(k)$. Bottom: $\chi_{sh} = 2$ holds for avalanche durations $-0.3$–2 s for $k = 13 \pm 4$ (mean ± SD over 8 mice). **c** Peak in pattern

complexity ($Cx$) with temporal coarse-graining (pattern depth $D = 5$, thresholded population activity). Top: each experimental condition. Bottom: E/I-model. With temporal coarse-graining the pattern complexity peaks in subsampled, critical dynamics ($g = 3.5$; $f = 0.1\%$, $\Theta = 1$), but monotonically decreases for subsampled, subcritical dynamics ($g = 3.75$); pattern depth $D = 5$. For each condition and model, mean complexity is presented. Error bars are the SD across 10 equal sections of each experiment/simulation. **d** Temporal coarse-graining at which $\chi_{sh} \geq 2$ maximizes pattern complexity (top) and temporal correlations (bottom). Averages over all experimental conditions and model. Error bars indicate range of $k$ for which $\chi_{sh} \geq 2$ and 90–100% of maximal $AC(1)$ or $Cx$, at baseline of $k = 1$ (Pearson' r = 0.53 for $Cx$ vs. $k$; Pearson' r = 0.85 for $AC(1)$ vs. $k$; $n = 6$). Broken line: slope of 1. For color code see (**c**).

both ongoing and evoked activity independent of pattern depth (Fig. 4c, d; Supplementary Fig. 15a, b). Our experimental findings were confirmed in the E/I-balanced model but only for critical dynamics (Fig. 4c, bottom), whereas subcritical dynamics lacked peak complexity with temporal coarse-graining. Our simulations also demonstrated that lower sampling fractions require higher temporal coarse-graining to recover peak complexity (Supplementary Fig. 15c), which is in line with our findings for $\chi_{sh}$ in the data (*cf.* Fig. 3e). This increase in temporal complexity was in line with our finding that the delayed auto-correlation AC(1) peaked for parabolic avalanches at the $k$ for which $\chi_{sh} = 2$, which was not found for flat avalanches or when all avalanches were taken into account (Fig. 4d; Supplementary Fig. 16).

Theory and experiment predict that the slope value of $\chi_{sh} \cong 2$, consistently found in our experiments, predicts scale-invariant, inverted parabolic profiles for avalanches, which can be collapsed with

an exponent $\chi^{coll} \approx 2$ (refs. 20–22; see "Methods"). We show in Fig. 5 the corresponding collapsed avalanche profiles for few- and many-generation avalanches for all experimental conditions and the critical model. Indeed, an inverted parabolic shape was only found for temporal coarse-graining $k$ that maximized $\chi_{sh}$ (Fig. 5a, top) for short-lasting, i.e., few-generation avalanches but not long-lasting, many-generation avalanches (Fig. 5a, bottom). The corresponding value of $\chi^{coll}$ was significantly increased to approximately 2 after temporal coarse-graining of few-, but not many-generation avalanches (Fig. 5b). Accordingly, after temporal coarse-graining to $\chi_{sh} \cong 2$, short-generation avalanches demonstrated a significantly better parabolic fit than many-generation avalanches, which revealed a flattened profile that deviated from a parabola (Fig. 5c). We note that parabolic avalanches ($L = 3–6$; $\chi_{sh} \geq 2$) did not recur regularly, which separates them from oscillatory activity (Supplementary Fig. 17). Using the temporally

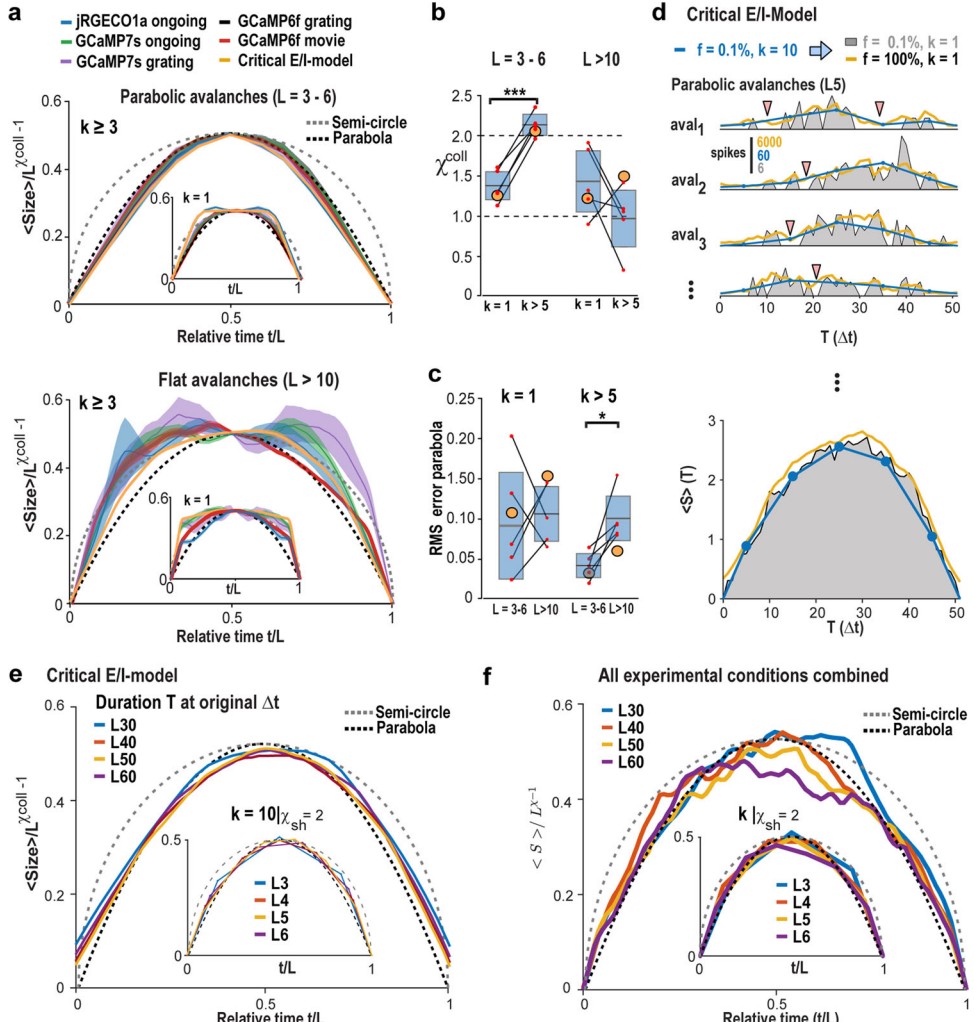

**Fig. 5 | Spontaneous and evoked neuronal assemblies follow the universal temporal profile of an inverted avalanche parabola with scaling collapse of $\chi = 2$. a** Top: for few-generation avalanches ($L = 3–6$; $k > 5$), inverted parabolas are recovered from flattened profiles present at $k = 1$ (inset). Bottom: Many-generation avalanches ($L = 10–20$) exhibit non-parabolic profiles at $k = 1$ (inset) and after temporal coarse-graining. Average profile ± SD. Avalanches were pooled from all recordings and mice per condition (color). Mean activity within an avalanche normalized by the number of generations $L$ to the power of $\chi^{coll}$-1 plotted for each relative time step, $t/L$. Note the match for the critical, subsampled E/I-model ($f = 0.1\%$, $\Theta = 1$), in line with the recovery of the critical exponent $\chi^{coll} = 2$, and inverted-parabola collapse for few-generation avalanches. **b** For few-generation avalanches, the scaling exponent for profile collapse, $\chi^{coll}$, is significantly higher and close to 2 after coarse-graining compared to $k = 1$, whereas no difference is found

for many-generation avalanches ($t = -11.94$, $D_F = 4$, $p = 0.00028$ vs. $t = 1.6$, $D_F = 4$, $p = 0.185$, two-sided paired t-test). Summary statistics for all conditions from (**a**). Orange circles: Critical E/I-model (SD < symbol size). **c** After temporal coarse-graining, short-lasting avalanches exhibit profiles closer to a parabola compared to long-lasting avalanches, but not at $k = 1$ ($t = -3.7$, $D_F = 4$, $p = 0.021$ vs. $t = -0.25$, $D_F = 4$, $p = 0.81$, two-sided paired t-test). **d** In the critical model, population activity sequences $t_i$ defined by $L = 5$ at $k = 10$ (blue) exhibit interruption of contiguous suprathreshold epochs due to subsampling (gray), not found in the fully sampled model (yellow). Bottom: Parabolic profiles of $L = 5$, $k = 10$ epochs (blue) match corresponding time averages from subsampled and fully sampled condition. **e** Summary for L30, L40, L50, and L60 at $\Delta t$ and corresponding avalanche profiles at $k = 10$ and $\chi^{coll} = 2$. **f** Corresponding analysis pooled over all experiments. **d–f** Averages are shown for clarity. Box plots: mean ± SD.

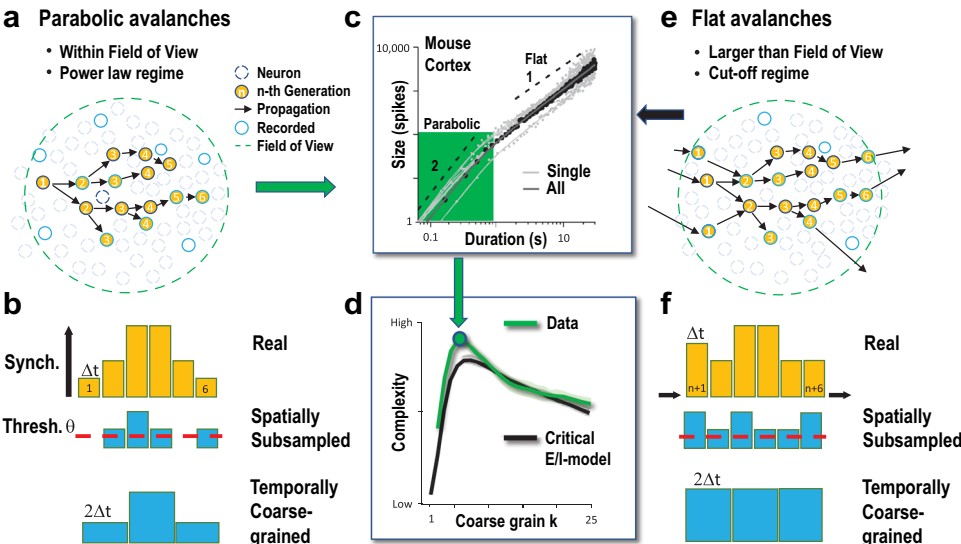

**Fig. 6 | Parabolic avalanches identify properly recovered synchronized cell assemblies in subsampled critical networks and exhibit high temporal complexity. a, b** Parabolic avalanches are constrained within the field of view and reside within the power law regime of the measured avalanche size and duration distributions. Despite the constraint of spatial subsampling, their temporal profile of synchronized neuronal activity can be recovered using thresholding and temporal coarse-graining. **c** Parabolic avalanches grow quadratically in size with duration. **d** The sequence of parabolic avalanches in the network exhibits maximal temporal complexity. **e, f** Flat avalanches are spatially larger than the field of view preventing a proper recovering of their time course using coarse-graining. This results in a flat average temporal profile and corresponding near linear growth with duration (see **c**, 1). Flat avalanches exhibit size and duration that lies beyond the cut-off of the size and duration distribution. For details see also Figs. 1, 4.

coarse-grained time segments of parabolic avalanches, we recovered their corresponding inverted parabolic shape in the average at the original temporal resolution of $\Delta t$ for the model and our data (Fig. 5d–f; Supplementary Fig. 18a, b). In the model, this approach can be used to recover segments of parabolic and flat avalanches in the subsampled system at original temporal resolution (Supplementary Fig. 18c).

A summary of our findings is shown in Fig. 6. In short, parabolic avalanches, in contrast to flat avalanches, can be properly recovered in subsampled critical networks within the field of view using temporal coarse-graining in combination with thresholding. We conclude that parabolic avalanches represent spontaneous and evoked synchronized cell assemblies in the cerebral cortex that exhibit high temporal complexity.

## Discussion

We identified an inverted parabolic profile in the synchronization of cell assemblies in prefrontal and primary sensory cortex. The symmetrical profile exhibits scale-invariance over hundreds of milliseconds to many seconds in line with predictions for parabolic avalanches. The corresponding temporal occurrences of parabolic avalanches maximizes the temporal complexity of neuronal synchronization during quiet resting and spontaneous locomotion as well as for visually evoked responses under a variety of different recording conditions.

The synchronized groups reconstructed here at cellular level for superficial layer 2/3 reside robustly within the framework of neuronal avalanches, defined by power laws in neuronal group size and duration within the experimentally defined field of view. These power laws, when interpreted within the framework of self-organized criticality (SOC; ref. 40), suggest a critical state of cortex that supports diverse and potentially system-wide propagation of synchronized activity[13,16,41]. This interpretation has been challenged recently by alternative models of avalanche generation utilizing balanced noise in the absence of neuronal interactions[23,42] and noise-induced fractures near a discontinuous, 1st-order phase transition including hysteresis[43], recently summarized as self-organized bi-stability (SOB; refs. 24,44). Our finding of a scaling exponent of $\chi = 2$ that is sensitive to spatial correlations

does not support these models, for which $\chi$ ranges between 1 and 1.5 and for which shapes are non-parabolic, closer to semi-circles or sawtooth like (e.g., refs. 23,24,34). Our results also clarify recent experimental findings on neuronal avalanches in cortex with a scaling exponent $\chi$ close to -1.3 (refs. 17,19,29). By introducing temporal coarse-graining in combination with deep-interpolation-based denoising, we overcame experimental limitations on spatial subsampling and demonstrated that indeed neuronal avalanches in cortex reflect synchronized cell assemblies exhibiting a scaling exponent of $\chi = 2$.

Our findings are in line with avalanche-generating mechanisms that reside close to a 2nd-order phase transition, belonging to the directed percolation universality class, e.g., a critical branching process (see also Fig. 3)[7,10,45–48] and specific types of processes found in intermittent magnetization near a critical state[21]. Our results are also in line with experimental results on 'Barkhausen noise' by Sethna et al.[21], who demonstrated power laws in size and duration with $\alpha \approx 1.5$ and $\beta \approx 2$, respectively, and an inverted parabolic profile with $\chi = 2$ for short avalanches, whereas a flattened profile and $\chi = 1$ was found for long avalanches[20] in line with recent findings in simulations of critical networks[27].

Our simulations also show that the recovery of $\chi = 2$ under subsampling conditions is only possible at criticality, regardless of coarse-graining level or threshold applied. While it has been shown that, under conditions of complete 'separation of time scales', subsampling in critical models can be compensated for by simply temporally integrating up to the finite-size cut-off of the system[49], these assumptions are not applicable for experimental data where the precise beginning and end of avalanches is unknown. Our simulations, though, demonstrate that temporal coarse-graining can recover $\chi = 2$ even when a 'separation of times scales' cannot be assumed. Extending beyond cortex, recent whole-brain zebra fish analysis demonstrated 'crackling noise'[21] to describe 3-dimensional propagation of activity with exponents $\alpha \approx 3$, $\beta \approx 2$, much steeper than in the current analysis, and estimates of $\beta$ to range between 1.6 and 1.8 (ref. 50).

It can be shown that the critical power law exponents for avalanche duration, size, and duration-size scaling are related by the formula $(\beta-1)/(\alpha-1) = \chi$ at the critical point, which was found for certain

models[21,22]. We consistently found that temporal coarse-graining preserves the power laws in avalanche size and duration distributions under subsampling conditions, however, it does so with increasingly shallower slopes[14] introducing a singularity in $(\beta-1)/(\alpha-1)$ when $\alpha=1$ (cf. Figs. 1f, 3c; Supplementary Fig. 8g). This singularity was even present in our subsampled model, despite the fully sampled model exhibiting critical exponents $\alpha=3/2$, $\beta=2$ which fulfilled the analytical prediction of $\chi=(\beta-1)/(\alpha-1)=2$. We conclude that spatial subsampling, as shown here and reported previously[51] as well as finite-size effects from spatial windowing[52] prevent straight-forward ratio calculations of the critical exponent $\chi$ from slope ratios obtained for size and duration distributions. We also note that this ratio relationship was derived in the absence of any external drive and with infinite separation of time scales, which differs from brain activity.

The coincident firing of neurons within a temporal window $\Delta t$ was used to identify brief periods of synchronous population activity. Our minimal threshold increased with temporal coarse-graining, which equates to requiring an increasingly higher minimal number of coincident spikes within the population of neurons. This synchronization requirement applies to each generation or $k \cdot \Delta t$ and thus, the overall duration of an avalanche does not affect the definition of minimal coincident spiking activity. Secondly, our randomization controls demonstrated that $\chi_{sh}$ rapidly drops below 2 and cannot be recovered by temporal coarse-graining. In these controls, we reduced the percentage of coincident spikes by either adding random spikes or removing correlated pairs of neurons for which correlation was measured at zero-time lag, thus quantifying coincident firing. Accordingly, we conclude that parabolic avalanches capture spiking activity of synchronized neuronal groups, i.e., cell assemblies, in the cortex.

The scale-invariant, inverted parabola represents a robust form of synchronization that complements commonly considered synchronization dynamics such as oscillations, waves, and synfire chains. The temporal profile of oscillations, while similar to an inverted parabola, would exhibit dominant durations and repeat regularly. Both aspects are not supported by our findings, which suggest scale-invariance without prominent periodicity. Propagating wave fronts that traverse the field of view would result in more flattened profiles and $\chi \approx 1$ like what we found for many-generation avalanches. A similar argument holds for synfire chains with their typically assumed constant layer width, which approximates extent-limited propagating wave fronts. Waves or synfire chains limited to the field of view would not be expected to unfold according to a scale-invariant, inverted-parabola.

Our finding of a symmetrical profile across numerous experimental conditions and spatial resolutions such as cellular spiking and the mesoscale-based LFP[28] suggests profile symmetry to be an important constraint for theories on brain synchronization. Such robust symmetry is unexpected. Avalanche propagation in the brain exhibits robust functional connectivity that is heavy-tailed or small-world[53,54], which simulations and theory suggest to support asymmetric profiles of critical network cascades[11]. Asymmetric avalanche profiles have also been reported for Barkhausen noise[55] when experimentally applying external forces, which, again would predict asymmetric profiles for, e.g., sensory-driven, neuronal avalanches.

The reconstruction of local synchronization reported here from cellular 2PI data approximates local synchronized activity captured in the local field potential[12,14]. Accordingly, our identification of $\chi=2$ for $450 \times 450$ µm field of view of superficial cortex using 2PI is in line with our recent demonstration of $\chi=2$ for LFP avalanches in superficial layers of nonhuman primates over an area that is 100 times larger[28].

We propose that the scale-invariant, inverted parabola in synchronization complements alternative frameworks of synchronization and potentially circumvents challenges presented by other measures. It is now well established that neuronal avalanches and oscillations co-exist in vitro and in vivo[28,56,57], a co-emergence dependent on e.g., the E/I-balance[33,58–60]. However, oscillations emphasize phase-locked firing

among neurons, which limits the number of patterns that can be phase-coded per cycle[61,62]. Similarly, although the spatial unfolding of neuronal avalanches favors nearby spatial sites in the aggregate[36], the spatial compactness of traveling waves[3,63] might limit spatial selectivity and simultaneous occurrence of waves within a brain region[64]. The relationship between avalanches and synfire chains is less clear. Synfire chains, by recruiting specific groups of neurons in each feed-forward layer[5,6,9], in principle could be highly adaptive and selective. On the other hand, they are difficult to stably embed in recurrent networks. In contrast, neuronal avalanches represent selective neuronal participation in propagated synchrony that maximizes the information that can be stored in highly diverse avalanche patterns[15,16,65,66]. Our results and simulations demonstrate a temporal gestalt of highly variable cell assembly synchronization in line with predictions of critical dynamics in cortex.

## Methods

### Overview on data sets analyzed

To demonstrate the robustness of our scaling results, we used 6 different data sets and 4 different probes to monitor ongoing activity in superficial layers of frontal cortex (ACC/mPFC) and evoked activity in superficial layers of primary visual cortex.

**Ongoing activity in contralateral ACC/mPFC monitored with jRGECO1a and GCaMP7s.** Mice (C57BL/6; Jackson Laboratories; age >6 weeks) were injected with a viral construct to express either GCaMP7s or jRGECO1a in cortical neurons using the Syn promotor. Chronic 2PI started after >2 weeks in the contralateral ACC/mPFC at an estimated depth of ~150–300 µm using a microprism. Recordings were collected over the course of several days from $n=5$ mice (3 males, 2 females; age 8–20 weeks) with jRGECO1a expression ($n=17$ recordings; 30 min each) and $n=3$ mice (all females; age 8–12 weeks) with GCaMP7s expression ($n=27$ recordings; 30 min each). Recordings were conducted over the course of several weeks and analyzed separately for each mouse.

**Ongoing activity in somatosensory cortex using transgenic Thy1-mice expressing GCaMP6s.** C57BL/6J-Tg(Thy1-GCaMP6s)GP4.3Dkim/J mice were obtained from Jackson labs (https://www.jax.org/strain/024275) and bred inhouse with C57BL/6J mice (Jackson Laboratory). Under a reversed 12:12 h light/dark cycle with ad lib access to a running wheel, mice were group-housed until the day of surgery and single-housed thereafter. Recordings of 10 min of ongoing activity from $n=2$ mice (all females; age >10 weeks) were performed over an area of ~1 mm² of cortical superficial layers.

**Visually evoked activity in V1 monitored with GCaMP7s.** Mice were injected with a mixture of a viral construct to express GCaMP7s in pyramidal neurons using the CaMKII promotor. Chronic 2PI started after >2 weeks in identified V1 at a depth of ~150 µm in response to drifting gratings from $n=2$ mice (all females; age >8 weeks) over the course of several days.

**Allen Brain Observatory Visual Coding dataset monitored with GCaMP6f.** Two additional data sets on visually evoked V1 responses were analyzed from the publicly available Allen Brain Observatory Visual Coding dataset (https://observatory.brain-map.org/visualcoding). These data were collected using GCaMP6f in $n=8$ transgenic mice (Cux2-CreERT2-GCaMP6f AI94; 3 females, 5 males; age 10–14 weeks) passively viewing drifting gratings and natural movies, imaged at 275 µm depth in V1 (~25 min recordings). For analysis, we used the deconvolved time series, extracted from the 2-photon ΔF/F signal using an L0 regularization algorithm, made available through 'allensdk' by the Allen Institute. We limited our analysis to recordings from superficial layers in V1 that contained at least several

hundreds of neurons and total stimulation time of many minutes. Data corresponding to 'drifting gratings' and 'natural movie 3' stimulation conditions were picked out of a longer recording using the 'allensdk' 'stimulus_table' object. For stimulation conditions with quickly varying stimulus, the 'stimulus_table' object was used to extract the start/end time for the entire stimulation epoch.

## Animal surgery

All procedures were approved by the NIH Animal Care and Use Committee (ACUC) and experiments followed the NIH *Guide for the Care and Use of Laboratory Animals*. Mice were obtained from Jackson Labs, bred inhouse with C57BL/6 backgrounds (Jackson Laboratory) under a reversed 12:12 h light/dark cycle. Chronic 2PI imaging was performed using a head bar in combination with a cranial window implanted in adult (>6 weeks) mice. Implants consisted either of (1) a microprism/ coverslip assembly to image through the medial wall of the contralateral ACC/mPFC following the procedure detailed in ref. 31 or (2) a stack of circular glass cover slips using established protocols[67]. For ACC/mPFC recordings, 2–3 injections of virus (100–400 nL; <1 μL in total; $10^{13}$ vg/mL; pAAV/.Syn.NES-jRGECO1a.WPRESV40--AAV9, pGP-AAV-syn-jGCaMP7s-WPRE AAV9, Addgene) were administered into the hemisphere contralateral to the prism implant (+0.75–1.25 mm AP, 0.1–0.3 mm lateral, -250 μm below the pial surface). To monitor V1 activity, the cranial window was placed centered at -2.5 mm from the midline (right hemisphere) and -1 mm rostral to the lambdoid suture.

For mesoscope experiments using transgenic Thy1-GCaMP6s mice, adult mice (age >6 weeks) underwent a head bar surgery in combination with a slightly modified cranial window implant to allow for a larger area of imaging. In short, the window implant consisted of 3 layers of No. 0 coverslips. The top coverslip (5 mm diameter) was used to close the craniotomy, whereas two smaller diameter cover slips (4 mm diameter) were used to gently fill the cavity between dura and the removed skull in order to prevent bone regrowth. This procedure resulted in clear craniotomies over extended periods of time with an area of approximately 4 mm × 4 mm accessible for imaging. The cranial window was centered above the midline at -0 Bregma[68].

## Identification of V1 maps

Retinotopic maps of V1 and higher visual areas (HVAs) were generated for all mice prior to recording using published protocols[69,70]. Briefly, awake, head-fixed mice faced with their left eye a 19″ LCD monitor placed at 10 cm distance and tilted 30° towards the mouse's midline. Using Psychophysics toolbox[71], contrast-reversing, spherically corrected checkerboard bars were drifted across the screen vertically (altitude) and horizontally (azimuth) for each of the four directions (30 repeats per direction). Simultaneous wide-field imaging (Quantalux, Thorlabs) captured GCaMP7s fluorescence, which was averaged for each direction. Altitude and azimuth phase maps were calculated by phase-wrapping the first harmonics of the 1D Fourier transform for each of the four averages and subsequently subtracting the maps of the opposite directions[70]. Sign maps were generated by taking the sine of the angle between the gradients in the altitude and azimuth maps and processed[69]. Borders were drawn around visual area patches and overlaid onto anatomical reference images to identify V1.

## Visual stimulation and response measures

Visual stimuli were prepared in Matlab (Mathworks) using the Psychophysics Toolbox[71] and delivered via a monitor (Dell, 60 Hz refresh rate) placed -25 cm in front of the contra-lateral eye of the mouse. The stimulus was composed of moving gratings at 8 different directions presented for 1 s at maximum contrast, 0.04 cycles per degree and 2 cycles per s. Stimuli were interspaced by gray screen (average luminance matched to stimuli) for 7 s. Each direction was presented 20 times in randomized order, for a total of 160 iterations.

## 2PI imaging, pre-processing pipeline, and meta data collection

For standard 2PI, images were acquired by a scanning microscope (Bergamo II series, B248, Thorlabs Inc.) coupled to a pulsed femtosecond Ti:Sapphire 2-photon laser with dispersion compensation (Chameleon Discovery NX, Coherent Inc.). The microscope was controlled by ThorImageLS and ThorSync software (Thorlabs Inc.). The wavelength was tuned to either 940 nm or 1120 nm in order to excite GCaMP7s or jRGECO1a, respectively. Signals were collected through a 16× 0.8 NA microscope objective (Nikon). Emitted photons were collected through 525/50 nm (GCaMP7s) or 607/70 nm (jRGECO1a) band filters using GaAsP photomultiplier tubes. The field of view was ~450 × 450 μm. Imaging frames of 512 × 512 pixels were acquired at 45.527 Hz by bidirectional scanning of a 12 kHz Galvo-resonant scanner. Beam turnarounds at the edges of the image were blanked with a Pockels cell. The average power for imaging was <70 mW, measured at the sample.

For mesoscope imaging, images were acquired by a dual-plane 2 Photon Random Access Mesoscope (2P-RAM, Thorlabs Inc.) coupled to a pulsed femtosecond Ti:Sapphire 2-photon laser (Chameleon Discovery NX, Coherent Inc.). The Mesoscope was controlled by ScanImage software (ScanImage, Vidrio Technologies). The wavelength was tuned to 920 nm in order to excite GCaMP6s. The specimen was excited at NA = 0.6 and 2-photon signals were collected at NA = 1 using 4 separate GaAsP photomultiplier tubes for 2 channels and 2 planes of imaging. The field of view was -1 × 1 mm². Imaging frames of 1024 × 1024 pixels (yields to 0.97 μm per pixel in lateral direction) were acquired at -10.5 Hz by a 12 kHz resonant scanner combined with a virtually conjugated Galvo scanner set. Continuous recording times were limited to 10 min each. The total power delivered to the specimen for imaging was <70 mW, which was split approximately equally between the 2 planes. The dual-plane imaging adds a shadow of each plane onto the other. Redundant cells were removed by detecting cells located at the same coordinates (with a 2 μm tolerance) across the two planes that are highly correlated (>0.7) and then selecting only the one with the highest signal.

The obtained tif-movies in uint16 format were rigid motion-corrected via the python-based software package 'suite2p'[72]. Registered images were further denoised using machine-learning-based, deep interpolation[32] (see below) and then semi-automatically processed by suite2p for ROI selection and fluorescence signal extraction. For each labeled neuron, raw soma and neuropil fluorescence signals (red for jRGECO1a; green for GCaMP6f/s, GCaMP7s) over time were extracted for each ROI. Spiking probabilities were obtained from neuropil corrected fluorescence traces ($F_{corrected} = F_{ROI} – 0.7*F_{neuropil}$) via MLspike (https://github.com/MLspike) by utilizing its autocalibration feature to obtain unitary spike event amplitude, decay time, and channel noise for individual ROIs.

**Deep-interpolation.** Deep-interpolation[32] (Deep-IP; https://github. com/AllenInstitute/deepinterpolation) removes independent noise by using local spatiotemporal data across a noisy image stack of $N_{pre} + N_{post}$ frames to predict, or interpolate, pixel intensity values throughout a single withheld central frame. The deep neural network is a nonlinear interpolation model based on a UNet inspired encoder-decoder architecture with 2D convolutional layers where training and validation are performed on noisy images without the need for ground truth data.

After rigid motion correction, individual denoised frames were obtained by streaming one 60-frame ($N_{pre} = N_{post} = 30$ frames) registered, image stack through the provided Ai-93 pretrained model[32] for each frame to be interpolated. At an imaging rate of -45 Hz, these 60 frames correspond to a combined -1.3 s of data surrounding the frame to be interpolated. To study the effect of Deep-IP, we omitted this step in our analysis of ongoing activity from our jRGECO1a recordings. In

addition, we compared the additional effect of removing weakly population-correlated ROIs (see Supplementary Fig. 2).

**Locomotion speed.** During imaging sessions, mice were head-fixed on a wheel on which they were free to run during collection of ongoing activity. The wheel was arrested during visual stimulation to reduce trial-by-trial variability. Locomotion speed was recorded via a custom-made photo diode sensor attached to the bottom of the recording platform. As the mice ran, the wheel turning made its teeth periodically block the IR light emitted by the sensor. This produced a square-wave pattern of reflected IR light on the sensor. Using wavelet denoising and thresholding, an instantaneous locomotion speed was reconstructed at frame rate resolution. Population activity was correlated with locomotion for ongoing activity using instantaneous or 1-s smooth locomotion speed estimates. Cross-correlation functions were detrended, combined over recordings for each mouse, and averaged across mice.

## Postprocessing pipeline

Our postprocessing pipelines were custom-written in Matlab (Mathworks) and Python (www.python.org). Some routines utilized NumPy (https://numpy.org/) and Matplotlib (https://matplotlib.org/).

**Uncorrelated cell removal.** In order to focus on synchronized population activity, we identified those cells in the population that were uncorrelated with the overall neuronal population. This procedure proved necessary in data sets that were not denoised by deep-interpolation. In contrast, once our imaging data were denoised, the percentage of uncorrelated cells dropped to negligible values (see Supplementary Fig. 2).

For each cell $i$ in an event raster, the cell's population correlation $c_i$ was computed by taking the Pearson's R between the cell's activity, $r_i(t)$, and the all-except-$i$ population activity time series $p(t) = \sum_{j \neq i} r_j(t)$. Then, a null population correlation $c'_{i,\tau}$ was computed as the Pearson's R between the all-except-$i$ population activity and $r'_{i,\tau}(t)$, where $r'_{i,\tau}(t)$ is cell $i$'s activity time series shifted forward in time by $\tau$ frames, where the last $\tau$ frames are shifted to the start of the activity (circular shift). Null correlations were computed for $\tau$ in range $\tau \in [-100, 100]$. From this, a distribution of null correlations $C'$ was formed, and from that a z-scored correlation value was calculated as $z_i = (c_i - \bar{C'})/\sigma^2_{C'}$. All cells with $z_i < 0.01$ were removed from the recording.

**Continuous epochs of suprathreshold population activity.** Continuous periods of population activity were identified by applying a threshold $\Theta$ on the population activity $p(t)$, the sum of the spike densities from all neuronal ROIs at a given time $t$, such that:

$$p_\Theta(t) = \begin{cases} p(t), & p(t) > \Theta \\ 0, & p(t) \leq \Theta \end{cases} \tag{1}$$

This procedure is known as hard-thresholding and was employed for all analysis unless otherwise stated. See Supplementary Notes and Supplementary Fig. 6 for a comparison between hard-thresholding and soft-thresholding. For a given recording $p(t)$ and coarse-graining value $k$, the dependence of the number $N$ of epochs on the threshold $\Theta$, $N(\Theta)$ was obtained for a range of thresholds $\Theta \in [\Theta_1, \Theta_2]$ such that $\Theta_1$ was low enough that it removed no population activity from the time course and $\Theta_2$ was high enough that it would remove all population activity from the time course. The function $N(\Theta)$ was typically well-approximated by a log-normal distribution and a corresponding fit yielded shape parameters $\mu$ and $\sigma$. The threshold used in the analysis for all recordings and coarse-graining factor was chosen such that $\Theta = \mu - 2\sigma$ and estimated for each $k$.

We note that thresholding the population activity obtained from a set of spiking neurons was first applied by Poil et al.[33]. in neuronal simulations to study neuronal avalanches. This method is similar to thresholding the local LFP (see, e.g., ref. 14), the traditional approach in population-based analysis of neuronal avalanches, where it was shown that the amplitude of the local negative deflection in the LFP monotonically increases with the number of neurons firing within the vicinity of the microelectrode (e.g., ref. 12). Importantly, we chose $\Theta$ to be low so as to minimize potential errors in the estimate of $\chi^{34}$ and thereby reduce the number of epochs obtained for our scaling analysis.

**Temporal coarse-graining.** A temporal coarse-graining operation was applied to the thresholded population activity $p_\Theta(t)$. For a given temporal coarse-graining factor $k$ an ensemble of $K$ different coarse-grained time series $p_k^0(\tau), p_k^1(\tau), \ldots, p_k^{K-1}(\tau)$ was arrived at through the following method:

$$p_k^j(\tau) = \sum_{i = k\tau + j}^{k(\tau+1)+j-1} p_\Theta(i) \text{ for } \tau \in \{0, 1, \ldots, \lfloor (T-j)/k \rfloor\} \tag{2}$$

For each time series $p_k^j(\tau)$, epochs were extracted by finding pairs $(\tau_1, \tau_2)$ such that $p_k^j(\tau_1) = 0$, $p_k^j(\tau_2) = 0$ and $p_k^j(\tau') > 0$ for all $\tau' \in \{\tau_1 + 1, \ldots, \tau_2 - 1\}$. The size of the epoch is given by $S = \sum_{i=\tau_1}^{\tau_2} p_k^j(i)$ and its corresponding duration given by $\tau_2 - \tau_1 - 1$. For a given ensemble of coarse-grained time series, all epochs were combined.

**Scaling curve fit.** For more precise evaluation of $\chi_{sh}$ and $\chi_{lg}$, we introduced the following fitting function:

$$S(d) = \frac{Cd^{\chi_{sh}}}{\left(1 + (d/\Phi)^\gamma\right)^{(\chi_{sh} + \chi_{lg})/\gamma}} \tag{3}$$

This function is a double power law with initial slope $\chi_{sh}$, transitioning to a second slope $\chi_{lg}$ at around the point $d = \Phi$. The parameter $\gamma$ controls how abruptly that transition happens and has been fixed at 4 for all the curves presented. The other parameters were free to adjust to the data, and all fits were performed in log-space, i.e., the $\log(S(\log(d))$ was fit to the log of data (taking the log of both average sizes as well as durations).

**Removal of inter-neuronal correlations by temporal shuffling.** Removal of inter-neuronal correlations was obtained by circular temporal shifts $T$ of the entire time series of individual neurons. $T$ was chosen randomly between 0 and up to the full length of the recording. These circular shifts maintain the precise temporal organization of each neuronal time series. This approach isolates contributions from inter-neuronal correlations while maintaining the first-order statistics and sequences of inter spike intervals. When we tested for the contribution of inter-neuronal correlations, for each percentage of correlations removed, we averaged over $n = 10$ repeats (see Fig. 1j).

**Random spike addition analysis.** To analyze the influence of uncorrelated activity on scaling trends, random spikes were added to recorded rasters after uncorrelated cell removal. This was done by adding spikes to the raster randomly until the desired noise level was reached. We note that 100% noise addition corresponds to doubling the firing rate in the raster. For each percentage of spikes added, we averaged over $n = 10$ repeats.

**Cell removal analysis.** To analyze the robustness of scaling trends to cell removal, cells were removed from rasters progressively until the scaling trends were destroyed. ROIs were selected randomly and removed. Analyses were repeated 5 times and the scaling trends were averaged over.

**Trial shuffling.** Trial-shuffling for the GCaMP7s and GCaMP6f drifting grating evoked data was obtained by randomly permuting the responses from each of the presented directions separately. This was done for each neuron independently. Therefore, in each trial of the trial-shuffled data set activity from each cell corresponds to a response to the same stimulus presented in the original data, but taken from different presentations of that stimulus. Note that for shuffled GCaMP6f movies, we employed circular random shifts instead of movie repeats.

**Bootstrapping.** Robustness of avalanche scaling trends and temporal profiles was assessed using bootstrapping. For a given recording, avalanches were calculated with the normal pre-processing steps. The pool of avalanches was then resampled 10 times with replacement, such that each bootstrapped pool contained the same number of avalanches as the original pool. Error bars for each plot were reported as the standard deviation of the scaling trend or temporal profile across all bootstrapped pools.

**State dependency of $\chi_{sh}$.** To study the effect of behavior state on network statistics and avalanche dynamics, spike rasters for rest and locomotion periods were extracted using wheel traces from individual recordings as follows ($n = 3$ mice and $n = 6$ recordings for which resting or locomotion periods were >15% of total recording time to allow for within-recording comparison). First, the instantaneous wheel speed was binarized using a threshold of 0.25 cm/s. Then, using this vector as a mask, we obtained separate rest and locomotion spike rasters for each recording. Spike rate and pairwise correlation CDFs were computed (Supplementary Fig. 7a, b). Avalanches for rest and locomotion were extracted separately and mean avalanche size vs. duration, scaling exponents, number of avalanches, and thresholds were computed for different temporal coarse-graining values of $k$ (Supplementary Fig. 7c–f).

**Complexity analysis.** Complexity analysis was performed as described in ref. 39 exploring patterns of depths D in the range of 4–7. Complexity $C$ was calculated on the population activity as a function of $k$ and thresholding, identically to how epochs were computed to obtain scaling. Subthreshold activity was not evaluated. For each segment of length D, its pattern $p$ is defined as the rank order of the time series (e.g., for a monotonically increasing trace over $D = 4$ time points, the pattern would be 0123; see Supplementary Fig. 15 for more examples). From all possible length $D$ segments, a probability distribution $P \equiv \{p_j : j = 1, 2, \ldots, N\}$ is obtained ($N$ is the number of possible states, e.g., for $D = 3$, the possible states are 012, 021, 102, 120, 201, and 210). Next, the Shannon's logarithmic information is computed as

$$S[P] = -\sum_{j=1}^{N} p_j \ln(p_j), \tag{4}$$

and the normalized Shannon entropy is defined as $H[P] = S[P]/S[P_e]$, where $P_e$ is the uniform distribution (entropy is maximized for the uniform distribution, therefore $0 \leq H \leq 1$). Finally, the complexity measure is defined as

$$C[P] = Q_J[P, P_e] \times H[P], \text{ with } Q_J[P, P_e] = Q_0 \left( S\left[\frac{P + P_e}{2}\right] - \frac{S[P]}{2} - S[P_e]/2 \right). \tag{5}$$

$Q_O$ is a normalization constant ($0 \leq Q_J \leq 1$) equal to the inverse of the maximum possible value of $J[P, P_e]$.

**Temporal profiles.** Temporal profiles were calculated on individual recordings after thresholding and temporal coarse-graining had been applied. Avalanches were obtained as described for our experimental data and grouped based on duration. Population activity during all avalanches of a certain duration was averaged over all avalanches to get the temporal profile for that duration.

To calculate the profile collapse exponent, $\chi^{coll}$, temporal profiles were first x-rescaled to fit between 0 and 1, then linearly interpolated to $N = 500$ points. Every interpolated temporal profile in the desired duration range was then y-rescaled by $d^{\chi^{coll}-1}$. The exponent $\chi^{coll}$ was chosen to minimize the RMS error between all collapsed shapes in the desired duration range. To calculate temporal profiles at a certain coarse-graining $k$ using the original experimental frame rate, avalanche epochs were calculated at the coarse-grained resolution as described above. Then, these epochs were mapped back onto the raster at the original frame rate and temporal profiles were calculated on these remapped epochs. The standard deviations from the mean rescaled shapes were visualized as shaded area if not stated otherwise.

## Neural simulations

**Model topology.** We adapted the model by Girardi-Schappo and colleagues[73], an excitatory/inhibitory (E/I) balanced system of integrate-and-fire (IF) neurons, which exhibits 4 domains of firing (low rate irregular, high rate, quiescent and intermittent) in addition to a 2nd-order, continuous phase transition. At this phase transition, the model displays the dynamics of a critical branching process with a slope of $\alpha = 3/2$ in avalanche size distribution and $\beta = 2$ for avalanche duration distribution, which has been identified experimentally for neuronal avalanches[7]. Our standard network consisted of $N = 10^6$ non-leaky, probabilistic integrate-and-fire neurons. The E/I neuron ratio was set 4:1 to approximate prevalence of excitatory over inhibitory neurons in the cortex and neurons had an all-to-all connectivity. The elements of the connectivity matrix, W, were initialized as $W^{EE} = W^{IE} = J$; $W^{II} = W^{EI} = -gJ$, that is connections with an excitatory pre-synaptic neuron had synaptic strength J and synapses with an inhibitory pre-synaptic neuron had synaptic strength $-gJ$. Here, g is the synaptic balance parameter which was used to tune the model to an E/I-balanced state with critical dynamics, and its critical value for the parameters of our network is, $g_c = 3.5$ (ref. 73). Values of g higher than $g_c$ shift the network to an inhibition-dominated regime with subcritical dynamics, and values of g lower than $g_c$ shifts the network to an excitation-dominated regime with supercritical dynamics.

**Model dynamics.** The state of a neuron in the model was described by 2 variables. A Boolean variable, X, denotes a neuron's firing (or quiescence) at time $t$, i.e., $X(t) = 1$ (or $X(t) = 0$). The membrane potential, V, controls the probability of firing and evolves as

$$V_i(t+1) = \left(\mu V_i(t) + \sum_{j=1}^{N} W_{ij} X_j(t)\right)(1 - X_i(t)) \tag{6}$$

where $\mu$ is a leakage parameter, which in our simulations was set to 0. The term $(1 - X_i(t))$ introduces an absolute refractory period of $\Delta t$ as it resets the voltage after a spike, i.e., when $X_i(t) = 1$. The probability of extra neuronal firing increased linearly with V according to

$$P(X = 1|V) = \begin{cases} V, & \Gamma V < 1 \\ 1, & \Gamma V \geq 1 \end{cases} \tag{7}$$

where $\Gamma$ is the neuronal gain, which in our simulation was set to 1 supporting probabilistic firing near a 2nd-order phase transition[73]. This network is non-conservative with energy dissipating through inhibition and spike collision.

**Model simulations.** The mean-field activity of the neurons show the hallmarks of neuronal avalanches with $\alpha = 3/2$, $\beta = 2$, and $\chi = 2$ (ref. 73), which we confirmed in our finite-size, fully-sampled network (see also Supplementary Fig. 10). Activity in the network is triggered by

independent Poisson processes to each neuron at low rate set to activate ~20 neurons on average (0.002% of the network) every time step. This external driving triggers intermittent, reverberating activity in the network, which leads to a neuronal firing rate of ~2.5 spikes per 1000 time units for the critical state. 1 time unit in our model thus equates to about 2.5 ms in real time when approximating the ~1 Hz average neuronal firing in our data. However, it is currently not possible to match the model further with our in vivo data given their unknown sampling fraction f, their true spike rate, neuronal topology, and corresponding threshold dependence of k.

For our analysis, we thresholded the population activity above the expected average number of spikes from the external drive to focus on the actual cascading activity in the network. Unless stated otherwise, the network was simulated for $10^8$ time-steps and the resulting time-series was analyzed. Epochs were extracted as described for the data with identical size and duration definition. We consider a random fraction $f$ of the total neurons in the network. This sampling fraction, $f$, was systematically varied from 0.01% (100 neurons) all the way to a fully sampled network ($10^6$ neurons) with corresponding changes in $\Theta$ to study the effects of subsampling on the network activity. All curves of $\chi_{sh}$ and $\chi_{lg}$ for the simulated data use the scaling curve fit described above.

## Statistics

All values are given as mean ± standard deviation (SD) if not stated otherwise. Violin plots were used to approximate the distribution of data (shaded area) bordered by the 1st and 3rd quartile. Box plots are composed of the median, box (25–75% quartiles) and whiskers (1.5× interquartile range).

## Reporting summary

Further information on research design is available in the Nature Portfolio Reporting Summary linked to this article.

## Data availability

The preprocessed imaging data used in this study are available in the general repository Zenodo using the following access https://doi.org/10.5281/zenodo.7703224 (ref. 74). The source data for all figures and supplemental figures in this study are provided for this paper. Source data are provided with this paper.

## Code availability

Computer code used in this study is available at https://github.com/PlenzLab/ParabolicAvalanches/.

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

## Acknowledgements

We thank Drs. James P. Sethna and Lucia de Arcangelis for discussions on scaling relationships between critical exponents. We thank Craig C. Stewart, Amber J. Tietgens for help with animal surgery, animal care and anatomical reconstructions and the NICHD Microscopy and Imaging Core (MIC) for help with rodent perfusion. This research was supported

by the Division of the Intramural Research Program (DIRP) of the National Institute of Mental Health (NIMH), USA, ZIAMH002797 (D.P.), ZIAMH002971 (D.P.), and the BRAIN initiative Grant U19 NS107464-01 (D.P., D.R.C.). This research utilized the supercomputing resources of the National Institutes of Health (NIH, USA; Biowulf, http://hpc.nih.gov) and the University of Maryland (UMD College Park, USA, https://hpcc.umd.edu/hpcc/dt2.html).

## Author contributions

D.P. conceived and planned the study; P.K. and T.L.R. performed experiments; E.C., T.L.R., and D.P. took the lead in data analysis to which S.R.M., M.V., E.G., S.P., A.V., and D.R.C. contributed. K.S. took the lead in the simulations. P.K. took the lead on deep-interpolation analysis. All authors contributed to experimental aspects, discussion, and writing of the manuscript.

## Funding

## Competing interests

The authors declare no competing interests.
