## [Peer Review File · Nature Communications]

Parabolic avalanche scaling in the synchronization of cortical cell assembliesReviewer #1 (Remarks to the Author):

Capek et al. recorded ongoing single-cell activity using 2-photon imaging (2PI) in the frontal and visual cortices of 8 mice during resting or spontaneously moving around. In 5 mice, 2PI was recorded in V1 in response to drifting gratings. The authors carefully check the dependence of avalanche size and duration scaling as a function of thresholding and temporal coarse graining and find a regime of avalanche durations and coarse-graining parameters where synchronized assemblies organize as scale-invariant avalanches that quadratically grow with duration. This corresponds to synchronous events lasting 0.2–2 s. Importantly, the authors reproduce similar relationships in a computational model that allows studying the effect of sub-sampling, indicating that the $X_{sh} = 2$ scaling is obtained for a wide range of sub-sampling conditions only when networks operate in the E/I-balanced critical regime. Finally, the authors find that pattern complexity is maximized for the temporal coarse graining that gives the $X_{sh} = 2$ both in empirical and model data, suggesting that rapid synchronization is facilitated by criticality.

Overall, the paper is well-written, the data and their analysis of high quality. The findings add to the accumulation of empirical evidence indicating that critical brain dynamics exist in many species, during different conditions, and at many levels of neuronal organization. As with other recent papers in the field, the study shows the importance of carefully refined analysis in order for the analysis to confirm theory. In this case, the use of a deep-interpolation method to improve signal-to-noise ratio of the analyzed data. As such, the study should inspire other groups to look at their data in ways that are not commonly practiced in neuroscience in general nor in the field of critical brain dynamics.

I have only minor comments that I would like the authors to consider when revising their manuscript:

1. I had some trouble with the problem definition (see comments #2–4 below). I think some reorganization of the 1st vs. 2nd paragraph would make the Introduction more approachable for the general reader as the scaling exponent X is introduced in the 2nd paragraph but implicitly referred to in the first paragraph.

2. The authors make a distinction between oscillations, waves, synfire chains and avalanches; however, these phenomena are at not necessarily different from one another in vivo. E.g., there are models showing oscillations composed of cycles of avalanches (Poil et al., 2012), which also seems to be the phenomenon reported by Peterman et al., (PNAS, 2009). Thus, I would suggest that the authors are less categorical or at least discuss to what extent these are different terms or different phenomena.

3. Related to point #2, I do not quite agree with the statement (L. 47–49): “In contrast to oscillations, waves and synfire chains, the spatiotemporal profile of neuronal avalanches has been difficult to predict theoretically and to identify experimentally”. Is the spatiotemporal profile of “oscillations, waves and synfire chains” well-known?

4. Similarly for the last sentence of the 1st paragraph: I do not find it clear what is meant with “robust unfolding of neuronal synchronization with a universal temporal profile similarly to oscillations, waves and synfire chains”. First of all, I do not feel that the reader is prepared at this stage to understand the meaning of “universal temporal profile”. Secondly, it is not clear from the references whether this universal profile is documented for oscillations, waves and synfire chains? If so, specific references after each entity would help.

5. For the Results, to prevent an artificial separation of the literature on similar signals/phenomena, I would appreciate conventional power spectra of the data (possibly as a supplementary figure). To offer an impression of the extent to which the data overall are oscillatory, or possibly scale-free, on various time scales.

6. Fig. 1f: please explain what is the meaning/interpretation of the singularity pointed out in the legend. Not clear from Results. There is some mentioning in Discussion but that is too late, and also in the Discussion the significance of this ratio could be better explained to prevent that the paper becomes too specialized or restricted in its audience.

7. Fig. 1 vs. 2: it is surprising that (or unclear why) the power-law distributions are shown with reference lines of $\alpha = 1$ and $\beta = 1$, when critical exponents of $3/2$ and 2 are expected. And it seems these exponents are produced by the model at no coarse graining? Please justify these choices or comment on why these conventional criteria are relaxed.

8. Fig. 2: I miss information about the time scales in the model. As I understood it, the time frames of the 2PI data were ~ 22 ms, making $K = 25$ correspond to bins of ~ 550 ms (which seems rather extreme?). What does $k = 30$ correspond to in the model? Are the absolute time scales comparable?

9. L. 157: I believe the authors meant to refer to Fig. 2g?

10. L. 182: I cannot quite follow the rationale here, isn't it almost trivial that X_{sh} will initially increase and subsequently decrease? Also, I'm not sure what I should understand with a "maximization principle in the temporal organization". Please rephrase or explain.

11. L. 198: "In contrast, ..." I think something went wrong in this sentence?

12. Complexity analysis: please provide more details. As a minimum a qualitative description of the measure.

14. Mice often fall asleep when they do not run around. We sleep periods omitted from the data?

Overall, a great paper that I learned a lot from.

Klaus Linkenkaer-Hansen

Reviewer #2 (Remarks to the Author):

This is an interesting paper, that explores the critical behavior of neuronal population activity in the superficial layers of the visual cortex and prefrontal cortex of the mouse, using 2-photon imaging. They do not do cell by cell analysis but compute aggregate activity which they coarse grain at different temporal scales and use this to estimate avalanche size and duration. They show that show that aggregate activity organizes as scale-invariant avalanches that quadratically grow in size as a function of their duration like an inverted parabola with collapse exponent 2 (for avalanches of absolute size 0.2 - 1.5 s over 0.2mm^2 of cortex). They show that this depended on spatial correlations. This pattern fits a certain percolation universality class that favors long range neuronal synchronization. They argue this is a universal scaling law of cell assembly synchronization that maximizes temporal complexity. At larger time scales this did not hold true as the collapse exponent dropped to 1 . Modelling confirmed that the temporal coarse graining process employed could extract this coefficient accurately from spatially sub-sampled networks. In the visual cortex, they demonstrate that the same $x=2$ scaling occurs during sensory processing.

The analysis is mathematically sound and meets the expected standards. One question that remains is whether the scaling observed is due to information coding properties of the cortex per se versus activity modulation due to behavioral parameters that may indicate differing brain states in the animal. Were the eyes/pupil size monitored? Were the animals locomoting or not? IF these parameters were measured it would be interesting to indicate whether or not they affected the analysis presented or the exponents calculated?

Also the paper would benefit from additional explanation in a number of places, in order to make it more appropriate for the background of the general readership of Nat Com. For example, a more intuitive explanation of the significance of the scaling of the mean avalanche size with duration and the exponent that collapses the temporal avalanche profile would be helpful. Technical talk could be made softer and more intuitive in a number of places.

Reviewer #3 (Remarks to the Author):

In their manuscript "Avalanche scaling in the synchronization of cortical cell assemblies," authors established a new temporal coarse-graining procedure. The authors claim that it can recover the scaling relationship between duration and size of avalanches in Ca⁺⁺ recordings from mice (with various fluorescent reporters). They test the method's usability on synthetic data from probabilistic integrate-and-fire networks with excitatory and inhibitory neurons.

The questions are attractive and can make a relevant story on an interface of physics and neuroscience. However, I am not yet convinced by the very sophisticated data analysis. The small ranges of parameters that bring desired results and many design decisions need a solid justification. The results are also somewhat overstated, which needs to be cleaned up to avoid misleading interpretations.

The authors established a tailored type of scaling and coarse-graining that needs to be tested in more examples to make sure when it can identify the true state of the system. It is a bit worrying that one must select a precise parameter value. For example, the authors found a rather narrow range of k . For the model, one can use arbitrary long simulation time and thus generate an abundance of data, which I expect should heal some of the scaling problems. Why is it reasonable that even for a model, only a range of k 's gives the expected scaling?

It seems that the methods developed might be related to the dynamic RG (which I am not a specialist in). Is there a connection? Such temporal coarse-graining should be grounded in a solid physical theory.

For the scaling χ_{sh} , the power-law fitting is performed based on 4 data points (4 different durations of the avalanches). It is a very small number. Can it be justified? Possibly in the models, one can identify the factors related to the location of transition between scaling. Possibly these factors will also influence (and be visible) in the real data are?

Why is it reasonable to have different shape-scaling for different durations?

Authors often refer to observed phenomena as "synchronization dynamics ." However, most avalanches (as given by the power-law distribution) are tiny and thus can hardly be called "synchronization ." At the same time, the large events have a temporal extent of hundreds of ms, up to 2 or more seconds, that can be called synchronous only in a particular situation. And they are not obeying the scaling relationship. Therefore, I feel the term is misleading and should be removed throughout the arguments.

The abstract states: "Our results identify universal scaling in the rapid and diverse synchronization of cortical cell assemblies in the form of neuronal avalanches." However, the paper demonstrates this scaling only for small events of duration, about 5% of maximal durations. Same with maximization of complexity (see below). I think the paper needs to be carefully revised to not overstate the obtained results (that on its own, if properly justified, can be interesting material).

An important study of avalanche shapes by Gleeson and Durrett (Nat. Comm 2017) shows that asymmetric shapes can be expected if the network has specific topological features (for example, a scale-free network). However, the scale-free topology can be quite realistic for the brain networks; thus, the symmetric parabolic shape may not be the most expected outcome.

A very long list of sophisticated pre-processing applied to the data seems to be

important. For example, the Deep-IP changes the data statistics a lot. This extensive pre-processing makes conclusions hard to validate and reproduce. Can the analysis be made directly on the Ca data?

Another option to justify the temporal coarse-graining procedure is to use it on already established data sets. For example, from the authors' lab: Beggs & Plenz 2003 data, or the later data with Woodrow Shew. What will happen with temporal coarse-graining and scaling there?

More technical questions:

The choice of threshold to be $\mu - 2 \sigma$ of $N(\theta)$ seems quite arbitrary, and it is hard to understand how it should change with k and meaning is. How much do the results depend on this choice? What if we select θ such that we will have a scaled number of avalanches? (from $k=1$ to $k=2$, for example, just 50% or 75% of them?)

Similarly, the authors write: "we chose θ to be low so as to minimize potential errors in the estimate of χ ." This seems to be hard to defend: I expect that θ influences differently the duration and size of the epochs; thus, it probably can control the χ . Is it possible to get a specific χ also for $k=1$ if choosing a specific value of θ ?

From the sketch of the size of population activity (Fig 1e), it seems like the size S also includes the area under the threshold. This was shown to cause misleading interpretations of the scaling exponents (Villegas et al. PRE 2019).

The complexity measure used is a bit special and was just recently introduced; possibly, it would help the reader if a bit more details on how it was defined were added to the methods.

As for the results related to this complexity: how should we interpret the observation that the k 's for scaling of shapes of small avalanches and the k for largest complexity are somewhat linearly dependent? (they are even not equal)

Maybe I overlooked something, but I do not see lines 187 – 190 confirmed in Fig 3d (if the dashed line would be put at $k_\chi = k_{\text{complexity}}$, it seems to not be where the points are).

Minor:

1. Continuity: the coarse-graining k (line 431, in manuscript with images) appears before the definition of this k (line 450)
2. For the images with α and β , as in inset of Figure 1f. it would be good to also show the χ (divergence will be out of the range, but it will be still informative)
3. It seems that k and K are used for the same thing. (449 and 450)

REVIEWER COMMENTS

Reviewer #1 (Remarks to the Author):

Capek et al. recorded ongoing single-cell activity using 2-photon imaging (2PI) in the frontal and visual cortices of 8 mice during resting or spontaneously moving around. In 5 mice, 2PI was recorded in V1 in response to drifting gratings. The authors carefully check the dependence of avalanche size and duration scaling as a function of thresholding and temporal coarse graining and find a regime of avalanche durations and coarse-graining parameters where synchronized assemblies organize as scale-invariant avalanches that quadratically grow with duration. This corresponds to synchronous events lasting 0.2–2 s. Importantly, the authors reproduce similar relationships in a computational model that allows studying the effect of sub-sampling, indicating that the $X_{sh} = 2$ scaling is obtained for a wide range of sub-sampling conditions only when networks operate in the E/I-balanced critical regime. Finally, the authors find that pattern complexity is maximized for the temporal coarse graining that gives the $X_{sh} = 2$ both in empirical and model data, suggesting that rapid synchronization is facilitated by criticality. Overall, the paper is well-written, the data and their analysis of high quality. The findings add to the accumulation of empirical evidence indicating that critical brain dynamics exist in many species, during different conditions, and at many levels of neuronal organization. As with other recent papers in the field, the study shows the importance of carefully refined analysis in order for the analysis to confirm theory. In this case, the use of a deep-interpolation method to improve signal-to-noise ratio of the analyzed data. As such, the study should inspire other groups to look at their data in ways that are not commonly practiced in neuroscience in general nor in the field of critical brain dynamics.

Response: We thank the reviewer for his succinct summary of our novel findings and their potential high impact for the field of neuronal synchronization.

I have only minor comments that I would like the authors to consider when revising their manuscript:

1. I had some trouble with the problem definition (see comments #2–4 below). I think some reorganization of the 1st vs. 2nd paragraph would make the Introduction more approachable for the general reader as the scaling exponent X is introduced in the 2nd paragraph but implicitly referred to in the first paragraph.

Response: We have now revised our Introduction to the general problem of synchronization in cascading activity specifically in avalanches at cellular resolution. We have expanded in the Discussion on differences between other forms of synchronization such as oscillations, waves, and synfire chains and now point out that these subcategories can co-exist with each other as pointed out in the reviewer's point #2 (Discussion, ~lines 350+)

2. The authors make a distinction between oscillations, waves, synfire chains and avalanches; however, these phenomena are not necessarily different from one another in vivo. E.g., there are models showing oscillations composed of cycles of avalanches (Poil et al., 2012), which also seems to be the phenomenon reported by Peterman et al., (PNAS, 2009). Thus, I would suggest that the authors are less categorical or at least discuss to what extent these are different terms or different phenomena.

Response: We agree with this reviewer and now include relevant literature on the co-existence of these categories in the Discussion (Discussion, ~lines 350+)

3. Related to point #2, I do not quite agree with the statement (L. 47–49): “In contrast to oscillations, waves and synfire chains, the spatiotemporal profile of neuronal avalanches has been difficult to predict theoretically and to identify experimentally”. Is the spatiotemporal profile of “oscillations, waves and synfire chains” well-known?

Response: We agree with the reviewer that the topic is complex and spatiotemporal profiles of oscillations, waves and synfire chains are much more diverse than as typically presented in the literature. We removed this sentence from the Introduction and expanded on this problem in our Discussion (lines ~325+)

4. Similarly for the last sentence of the 1st paragraph: I do not find it clear what is meant with “robust unfolding of neuronal synchronization with a universal temporal profile similarly to oscillations, waves and synfire chains”. First of all, I do not feel that the reader is prepared at this stage to understand the meaning of “universal temporal profile”. Secondly, it is not clear from the references whether this universal profile is documented for oscillations, waves and synfire chains? If so, specific references after each entity would help.

Response: We agree with the reviewer that this topic is presented too early in the manuscript. We have expressed this argument in more layman terms and have now avoid the term ‘universal’ and related theory-heavy terminology to the Discussion.

5. For the Results, to prevent an artificial separation of the literature on similar signals/phenomena, I would appreciate conventional power spectra of the data (possibly as a supplementary figure). To offer an impression of the extent to which the data overall are oscillatory, or possibly scale-free, on various time scales.

Response: Spike density estimates from fluorescence changes even at our relatively high frame rate of ~45 Hz typically cover a narrow frequency regime of 1 – 20 Hz. This is due to the removal of slow baseline changes required for spike density estimates, the decay constant of the reports (~1s), and our sampling frequency. In addition, suprathresholded population activity exhibits short epochs with disjunct transitions that create noisy contributions when calculating power spectra. In contrast, autocorrelations are more robustly calculated under these conditions and can selectively address short avalanches, our epochs of interest.

The referee might have missed the autocorrelations presented in Suppl. Fig. 13 of our original submission, now Suppl. Fig. 17 in revision. These autocorrelations demonstrate the absence of oscillatory components in the recordings which is to be expected given the very low sensitivity of 2PI recordings for oscillations. We point that these calculations were done for parabolic avalanches, which our work shows reflect properly resolved avalanches. We have also now include the recent simulation paper by the DeArcangelis group (Nandi et al. 2022) on the not so straightforward relationship between the exponents and 1/f scaling.

6. Fig. 1f: please explain what is the meaning/interpretation of the singularity pointed out in the legend. Not clear from Results. There is some mentioning in Discussion but that is too late, and also in the Discussion the significance of this ratio could be better explained to prevent that the paper becomes too specialized or restricted in its audience.

Response: We thank this reviewer to allow us to clarify this issue further. The unfortunate situation in the field is currently a conflation of critical exponents with slope estimates. It is a complex issue and would go way beyond the scope of our experimental findings. We have now removed this statement from the early portion of the paper and expanded on the significance of this ratio in more detail in the Discussion (~lines 299+).

7. Fig. 1 vs. 2: it is surprising that (or unclear why) the power-law distributions are shown with reference lines of $\alpha = 1$ and $\beta = 1$, when critical exponents of $3/2$ and 2 are expected. And it seems these exponents are produced by the model at no coarse graining? Please justify these choices or comment on why these conventional criteria are relaxed.

Response: We apologize for suggesting an importance of $\alpha = 1$ and $\beta = 1$ without a more detailed explanation. In fact, $k = 1$ is already an arbitrary scale introduced by the experimentalist and accordingly α and β slopes are not necessarily equivalent to critical exponents. These lines act as guide to the eye to help the reader realize that these slopes are relatively shallow and thus the distributions suggest highly variable sizes and durations, typical for avalanche activity.

As pointed out above, we have now explained the singularity in the ratio $(\alpha - 1)/(\beta - 1)$ and the fact that these estimates cannot be equated with critical exponents if subsampling is encountered in the Discussion.

The model produces the critical exponents $\alpha \sim 3/2$, $\beta \sim 2$ and $\chi \sim 2$ at full sampling (Fig. S10). We now point this out specifically in the Main text (line 185), in the Discussion (line 303+) and in the legend of previous Suppl. Fig. S6, now Suppl. Fig. S10.

8. Fig. 2: I miss information about the time scales in the model. As I understood it, the time frames of the 2PI data were ~ 22 ms, making $K = 25$ correspond to bins of ~ 550 ms (which seems rather extreme?). What does $k = 30$ correspond to in the model? Are the absolute time scales comparable?

Response: The average firing rate of neurons in our data is ~ 1 Hz (see Fig.1b), the typical firing rates of prefrontal cortex neurons. In the model, the external Poisson drive induces ~ 20 spikes/time unit. The resulting network activity in the critical state leads to a neuronal firing rate of ~ 2.5 spikes per 1,000 time units. Accordingly, 1 time unit in our model equates to about 2.5 ms in real time. We have added this information to the Material & Methods 'Model Simulation' (~line 706+). Thus, at $f = 0.1\%$, we recover $X_{sh} = 2$ at $k = 8$ and the temporal coarse-grained scale of the model would equate to $k \cdot dt = 8 \cdot 2.5$ ms = 20 ms. However, it is currently not possible to match the model further with our in vivo data given their unknown sampling fraction f , their true spike rate, neuronal topology, and corresponding threshold dependence of k .

In response to the reviewer's comment about $k=25$ (550ms) being extreme, we note that our least-noisy datasets reach $X_{sh}=2$ for $k_s < 5$; it is only for our most noisy data that extreme temporal integration is required to smooth out noise.

9. L. 157: I believe the authors meant to refer to Fig. 2g?

Response: Fixed.

10. L. 182: I cannot quite follow the rationale here, isn't it almost trivial that X_{sh} will initially increase and subsequently decrease? Also, I'm not sure what I should understand with a "maximization principle in the temporal organization". Please rephrase or explain.

Response: We have rephrased this paragraph accordingly (Results, ~lines 231+).

11. L. 198: "In contrast, ..." I think something went wrong in this sentence?

Response: Fixed.

12. Complexity analysis: please provide more details. As a minimum a qualitative description of the measure.

Response: We have now added more details in the Results section (~lines 232) and greatly expanded our corresponding M&M section ('Complexity Analysis').

14. Mice often fall asleep when they do not run around. Were sleep periods omitted from the data?

Response: State dependency of our results has been of great concern to us, which is why we monitored the locomotion of our mice on the wheel during ongoing activity. First, we would also like to point out that our nocturnal animals are housed under a reverse 12 hr light/12 hr dark cycle, which aligns their natural activity period with our imaging times. This should reduce the incidence of sleep periods during imaging. In addition, we have now categorized our ongoing activity recordings into quiet resting and locomotion (see M&M; new supplementary figure S7). We found no significant difference in X_{sh} between quiet resting and locomotion despite a . We conclude that sleep states, that are absent during locomotion, are not affecting our results. In visual stimulation experiments from our lab, we prevented wheel running to reduce trial-by-trial variability in these animals. If animals underwent brief periods of sleep, they did not affect our results, which were comparable with what was found for ongoing activity.

Overall, a great paper that I learned a lot from.

Klaus Linkenkaer-Hansen

Response: We thank you for your very thoughtful comments and thorough reading of our work.

+++++

Reviewer #2 (Remarks to the Author):

This is an interesting paper, that explores the critical behavior of neuronal population activity in the superficial layers of the visual cortex and prefrontal cortex of the mouse, using 2-photon imaging. They do not do cell by cell analysis but compute aggregate activity which they course grain at different temporal scales and use this to estimate avalanche size and duration. They show that show that aggregate activity organizes as scale-invariant avalanches that quadratically grow in size as a function of their duration like an inverted parabola with collapse exponent 2 (for avalanches of absolute size 0.2-1.5s over 0.2mm² of cortex). They show that this depended on spatial correlations. This pattern fits a certain percolation universality class that favors long range neuronal synchronization. They argue this is a universal scaling law of cell assembly synchronization that maximizes temporal complexity. At larger time scales this did not hold true as the collapse exponent dropped to 1. Modelling confirmed that the temporal course graining process employed could extract this coefficient accurately from spatially sub-sampled networks. In the visual cortex, they demonstrate that the same $x=2$ scaling occurs during sensory processing.

Response: We thank the reviewer for her/his succinct summary of our results and considering our results interesting in the context of system neuroscience and criticality.

1. The analysis is mathematically sound and meets the expected standards. One question that remains is whether the scaling observed is due to information coding properties of the cortex per se versus activity modulation due to behavioral parameters that may indicate differing brain states in the animal. Were the eyes/pupil size monitored? Were the animals locomoting or not? IF these parameters were measured it would be interesting to indicate whether or not they affected the analysis presented or the exponents calculated?

Response: We have now included new analysis by separating neuronal activity into periods when the animal was quiet resting vs. periods of locomotion or wheel running. There was no significant difference in our main results despite a significant higher average firing rate during locomotion (new supplemental figure S7). Please also see our response to Ref. 1, pt. 14.

2. Also the paper would benefit from additional explanation in a number of places, in order to make it more appropriate for the background of the general readership of Nat Com. For example, a more intuitive explanation of the significance of the scaling of the mean avalanche size with duration and the exponent that collapses the temporal avalanche profile would be helpful. Technical talk could be made softer and more intuitive in a number of places.

Response: We have now revised our Introduction to emphasize the significance of the scaling with respect to mean size and duration. We also reduced technical jargon given the general readership of Nat Communications.

+++++

Reviewer #3 (Remarks to the Author):

In their manuscript “Avalanche scaling in the synchronization of cortical cell assemblies,” authors established a new temporal coarse-graining procedure. The authors claim that it can recover the scaling relationship between duration and size of avalanches in Ca⁺⁺ recordings from mice (with various fluorescent reporters). They test the method’s usability on synthetic data from probabilistic integrate-and-fire networks with excitatory and inhibitory neurons.

The questions are attractive and can make a relevant story on an interface of physics and neuroscience. However, I am not yet convinced by the very sophisticated data analysis. The small ranges of parameters that bring desired results and many design decisions need a solid justification. The results are also somewhat overstated, which needs to be cleaned up to avoid misleading interpretations.

Response: We thank this reviewer for their assessment that our results ‘make a relevant story on the interface of physics and neuroscience.’ To clarify the robustness of our results and generality of our analysis pipeline, we have provided additional analysis and new data detailed below. We have also numbered the referee’s points of critique for simplifying cross referencing in the main text and between referees.

1. The authors established a tailored type of scaling and coarse-graining that needs to be tested in more examples to make sure when it can identify the true state of the system. It is a bit worrying that one must select a precise parameter value. For example, the authors found a rather narrow range of k . For the model, one can use arbitrary long simulation time and thus generate an abundance of data, which I expect should heal some of the scaling problems. Why is it reasonable that even for a model, only a range of k ’s gives the expected scaling?

Response: We have now added additional information about the range of k for which $X_{sh} = 2$ is observed.

For the model, we added supplemental figure S11 to show that the range of k is independent of the simulation length. The reason why X_{sh} decreases in the model is not due to limited simulation time, but rather due to the eventual concatenation of independently triggered avalanches. If we had employed an absolute separation of time scales, i.e. infinite inter-avalanche intervals, indeed X_{sh} would not drop for large k , and such a simplified model would exhibit $X_{sh} = 2$ beyond a certain k value. Instead of assuming infinite separation of time scales, we set our rate of external Poisson drives low as to avoid triggering new avalanches while a current avalanche unfolds. Accordingly, successive avalanches are still largely uncorrelated in the model, despite a finite average inter-avalanche interval. However, given our finite inter-avalanche interval, now avalanches become erroneously concatenated at large k and X_{sh} drops below 2. We have added a statement to the corresponding Results section (~lines 196 – 199).

For 2PI data, we now point out that long-duration avalanches for which scaling is not observed are primarily located in the cut-off of the size and duration distribution (Fig. 1f,g; jrGECO; Fig. 2a, Suppl. Fig. S8e, f; GCaMP7s; Fig. 3, model). It is well understood that the cut-off reflects improperly estimated avalanche activity due to finite-size effects (e.g. Yu et al., 2014). Accordingly, if k is too small, avalanches

are prematurely terminated (type II error) because spatial subsampling does miss those neurons that carried the avalanche further. If k is too large, avalanches including those in the cut-off are erroneously merged (type I error). Both errors lead to a scaling in mean avalanche size and duration that is much smaller than 2, which is close to the case of decorrelated activity as shown by our randomization controls (cf. Suppl. Figs. S2g, and S8h). These two errors are responsible for the bounded range in k for which $X_{sh} = 2$ is observed. We have added statement to the Results section (~lines 166 – 170).

We also note that removing noise from datasets broadens the parameter range at which $X_{sh} = 2$ (Fig S2.e), and removing correlations from a dataset sharpens the parameter range (Fig 1.j). This trend is seen across experimental conditions, where the largest FoV experiments (mesoscope, Fig 2h) and highest-fidelity indicators (jrGECO, Fig 1h) have broader k ranges than lower-fidelity indicator experiments (GCaMP6f, Fig 4b). This suggests that higher noise contributes to shortening critical k ranges. We further note that the other main parameter, the threshold, has a large range where $X_{sh} = 2$ (Fig S6c).

We believe that the reviewer's impression that we utilize 'very sophisticated data analysis' might have arisen from our original, terse description of our approach (see Results ~lines 98 – 110). Our analysis in fact constitutes two rather standard approaches in neuroscience. The first one is to sum spikes from many neurons into a population activity time series. The second one is the thresholding of this population activity to identify suprathreshold periods of interest. This last step has been employed by many groups and modelers to study avalanche dynamics in the brain. Our paper identifies crucial shortcomings in this approach when subsampling and noise is not considered. We show in this paper how to overcome these shortcomings and obtain a robust scaling exponent. We demonstrated the robustness of our scaling results for a wide variety of experimental conditions in neuroscience, i.e., ongoing vs. evoked activity, associative cortex vs. sensory cortex, locomotion vs. quiet resting (see below), and its dependence on correlations by employing advance shuffle-controls. In our model, we demonstrate that rescuing parabolic scaling using temporal coarse graining is limited to critical dynamics. We do not claim that temporal coarse graining will generalize to other types of models or provides a general avenue to recover critical exponents.

2. It seems that the methods developed might be related to the dynamic RG (which I am not a specialist in). Is there a connection? Such temporal coarse-graining should be grounded in a solid physical theory.

Response: RG would involve coarse-graining in both space and time, which indeed is being pursued by several groups regarding avalanche analysis in the brain. Our current approach, though, is much more data analysis driven as it addresses the specific aspects of failing to identify avalanches due to an incomplete observation and the presence of noise (see our comments on type I and type II error in pt. 1). As such, we do not claim that our findings require an underlying larger principle as identified with RG. We believe though that insights from our paper are an excellent starting point to study RG in this context.

3. For the scaling χ_{sh} , the power-law fitting is performed based on 4 data points (4 different durations of the avalanches). It is a very small number. Can it be justified? Possibly in the models, one can identify the factors related to the location of transition between scaling. Possibly these factors will also influence (and be visible) in the real data are?

Response: We thank the referee for pointing this out and accordingly, we have introduced additional data analysis steps and new data sets using a mesoscope for 2PI.

We have introduced a power law fit that quantifies the location of the transition in the scaling from $X_{sh} = 2$ to $X_{lg} \ll 2$ (See M&M 'Scaling curve fit'). We report the corresponding slope and transition values in the corresponding figures and legends.

Using a newly built mesoscope, we provide new data that the number of points in the power-law fit, the transition ϕ , increases by a factor of 2 – 3 when recording from a 4 times larger cortical area at similar temporal resolution (new Figs. 2e – h).

4. Why is it reasonable to have different shape-scaling for different durations?

Response: We apologize for not being clear regarding these subplots. In Fig. 4, we have provided corresponding waveform shapes at different k to demonstrate that only for the k at which X_{sh} is close to 2 we find $X_{collapse}$ to be 2 and the form to be a parabola. Similarly, we provide the deviations from a parabola for long-duration avalanches for which $X_{sh} \ll 2$ for any k .

The reason the shape-scaling is different for different durations is the same as the reason for why X is different for different durations.

5. Authors often refer to observed phenomena as “synchronization dynamics.” However, most avalanches (as given by the power-law distribution) are tiny and thus can hardly be called “synchronization.” At the same time, the large events have a temporal extent of hundreds of ms, up to 2 or more seconds, that can be called synchronous only in a particular situation. And they are not obeying the scaling relationship. Therefore, I feel the term is misleading and should be removed throughout the arguments.

Response: We believe that there is a misunderstanding as to the extent of synchronization that can be expected in normal cortical activity. It is well understood that neuronal firing in a recurrent, excitatory network like the cortex is tightly regulated by the balance of excitation and inhibition (see e.g. Renart et al 2010 Science, The asynchronous state in cortical circuits). Even a small reduction in inhibition readily leads to abnormally large, synchronous population events, known as seizures in epilepsy. Accordingly, our normal brain activity analyzed in this study reveals the low average pairwise correlation in the firing of neurons in vivo within the regime of $\sim 0.03 - 0.06$ (Fig. 1b), typical of healthy recordings in the awake state. On the other hand, it is well established that coincident firing between neurons robustly excites down-stream neurons which allows for neuronal activity to propagate in the network. Our approach utilizes this most fundamental definition of ‘synchronization’ among neurons in neuroscience: the coincident firing of spikes within a temporal window of width Δt .

We have now provided a more detailed explanation how the threshold applied to the population activity time course equates to requiring a minimal number of coincident spikes within the population of neurons. As our sketch in Fig. 1e reveals, this threshold increases with temporal coarse graining thus sharpening our coincident spike criterion. For the data, we show this increase in θ in Suppl. Figs. S3b, S5b, S7e. Importantly, because this coincidence criterion is applied at the original temporal resolution Δt ,

neither the overall duration of an avalanche, nor the temporal coarse graining factor k affects the definition of minimal synchronized activity.

Second, the randomization controls for our data (Fig.1j, 2b, d) demonstrate that X_{sh} rapidly drops below 2 and can not be recovered by temporal coarse graining. In these controls, we reduced the percentage of coincident spikes by either adding random spikes or removing correlated pairs of neurons (correlation measured at zero-time lag, quantifying coincident firing). In our model, we show the destructive nature of noise and how it can be partially compensated for by a higher threshold in a systematic manner (Suppl. Fig. S12). These controls clearly establish that our work centers on synchronized neuronal activity in cortex in the form of coincident neuronal firing. We have clarified this further by stating in the last sentence of the Abstract 'These results establish a novel scaling relationship for the coincident firing of cortical neurons in the form of parabolic avalanches.'

We thank the reviewer for this comment, which allowed us to clarify our position. We have added and revised language throughout the manuscript as to not imply that parabolic avalanches are the only synchronized activity in the population activity. In fact, by definition, any suprathreshold population activity within Δt , which is determined by our frame rate, is considered synchronized activity in our analysis and holds for parabolic avalanche periods as well as flat avalanche periods.

6. The abstract states: "Our results identify universal scaling in the rapid and diverse synchronization of cortical cell assemblies in the form of neuronal avalanches." However, the paper demonstrates this scaling only for small events of duration, about 5% of maximal durations. Same with maximization of complexity (see below). I think the paper needs to be carefully revised to not overstate the obtained results (that on its own, if properly justified, can be interesting material).

Response: We have removed the term 'universal' from the abstract and are now providing the maximal duration for which parabolic avalanches are observed. Please also see our responses to pts. 1 & 5 above.

7. An important study of avalanche shapes by Gleeson and Durrett (Nat. Comm 2017) shows that asymmetric shapes can be expected if the network has specific topological features (for example, a scale-free network). However, the scale-free topology can be quite realistic for the brain networks; thus, the symmetric parabolic shape may not be the most expected outcome.

Response: We thank this reviewer referring to this important aspect between topology and shape symmetry. We have now expanded in the Discussion on potential factors influencing avalanche shape symmetry and the novelty of finding experimentally robust symmetrical shapes (~lines 339 – 346).

8. A very long list of sophisticated pre-processing applied to the data seems to be important. For example, the Deep-IP changes the data statistics a lot. This extensive pre-processing makes conclusions hard to validate and reproduce. Can the analysis be made directly on the Ca data?

Response: All our pre-processing steps, except for the Deep-IP, are standard in the field of 2-photon imaging and highly preferred over Ca data. While Deep-IP improves the results, it is not required for the general result. For the Deep-IP processed jrGECO data set in Fig. 1f - k, the corresponding analysis without Deep-IP is presented in Suppl. Material (now Fig. S2e). Our GCaMP7s results (now Fig. 2a – d) represent recordings of ongoing activity without the Deep-IP processing step. This purely standard analysis also shows that it is the set of clearly identified synchronized neurons which exhibits this scaling

and its concomitant destruction by adding noise. For visually evoked data, the comparison with omitting Deep-IP has now moved to (Suppl. Fig. S14d – e). Our data analyses consistently show that $X_{sh} = 2$ is found at smaller k for better resolved data sets. In our model, we confirm this finding by artificially adding noise (Suppl. Fig. S12). Thus our analysis shows that deep-IP does not introduce the reported finding of $X_{sh} = 2$. Instead, our analysis demonstrates that X_{sh} will be missed if data sets are not of sufficient quality.

Because Ca data are highly smoothed by the $\sim 1s$ decay time constant of the fluorescent indicator, temporal coarse graining is difficult to interpret. In fact, we purposely utilize spike identification, which we consider the appropriate standard in the analysis of neuronal 2PI data at cellular resolution.

9. Another option to justify the temporal coarse-graining procedure is to use it on already established data sets. For example, from the authors' lab: Beggs & Plenz 2003 data, or the later data with Woodrow Shew. What will happen with temporal coarse-graining and scaling there?

Response: The data sets by Beggs & Plenz 2003 and subsequent work with Woodrow Shew were based on local field potential recordings in vitro. In fact, we recently published a scaling relationship of $X = 2$ for avalanches based on local field potential recordings in awake non-human primates (Miller et al., 2019; cited in our original submission). The technical advance of the present paper, though, lies in the demonstration of avalanche scaling and corresponding collapsed, symmetrical waveforms using cellular resolution 2PI data in vivo, which overcomes ambiguity in the interpretation of the origin of the LFP and is considered the current, high yard stick in systems neuroscience.

10. More technical questions:

a. The choice of threshold to be $\mu - 2 \sigma$ of $N(\theta)$ seems quite arbitrary, and it is hard to understand how it should change with k and meaning is. How much do the results depend on this choice? What if we select θ such that we will have a scaled number of avalanches? (from $k=1$ to $k=2$, for example, just 50% or 75% of them?) Similarly, the authors write: “we chose θ to be low so as to minimize potential errors in the estimate of χ .” This seems to be hard to defend: I expect that θ influences differently the duration and size of the epochs; thus, it probably can control the χ . Is it possible to get a specific χ also for $k=1$ if choosing a specific value of θ ? From the sketch of the size of population activity (Fig 1e), it seems like the size S also includes the area under the threshold. This was shown to cause misleading interpretations of the scaling exponents (Villegas et al. PRE 2019).

Response: We have added new analysis on the role of the threshold in obtaining X_{sh} with coarse graining. In Suppl. Information and Figure S6, we analytically derive the impact of the subthreshold regime on the scaling (‘Soft’ vs. ‘hard’ thresholding). We show that our results are robust to both approaches and provide the error bands for our ongoing activity data sets. We further show the robustness of $X_{sh} = 2$ to a large range of thresholds in corresponding colormaps for data and the critical model (Fig S6.e). We would like to point out that because of the large number of different data sets, experimental conditions (e.g. varying number of neurons recorded, duration of recordings, S/N ratio) used in the current manuscript, we opted for a standardized method to capture the dependency of total avalanches on θ . We found that these distributions were well fit by a lognormal, which supports the approach to choose θ consistently for all data sets and experimental conditions based

on the sigma of these distributions. We do not suggest that this is the only way to link an increase in threshold with k .

$X_{sh} = 2$ can not be reliably obtained at $k = 1$. The reason is that temporal coarse graining concatenates temporal 'gaps' introduced by spatial subsampling. These gaps are not corrected by just increasing the threshold. In fact, increasing the threshold also increases the number and duration of subthreshold epochs making it harder to obtain X_{sh} .

11. The complexity measure used is a bit special and was just recently introduced; possibly, it would help the reader if a bit more details on how it was defined were added to the methods.

Response: We have now expanded the corresponding sections in the manuscript. Please also see our response to Ref. 1, pt.12.

12. As for the results related to this complexity: how should we interpret the observation that the k 's for scaling of shapes of small avalanches and the k for largest complexity are somewhat linearly dependent? (they are even not equal). Maybe I overlooked something, but I do not see lines 187 – 190 confirmed in Fig 3d (if the dashed line would be put at $k_{\chi} = k_{\text{complexity}}$, it seems to not be where the points are).

Response: We do not claim that there is a linear dependency between avalanche shape and complexity. This scatter plot demonstrates that at the k at which we properly identify avalanches by compensating for spatial subsampling and noise with temporal integration, we preserve the highest amount of temporal correlations and complexity. We don't expect these k values to be equal given the large errors associated with their identification.

Minor:

1. Continuity: the coarse-graining k (line 431, in manuscript with images) appears before the definition of this k (line 450).

Fixed.

2. For the images with α and β , as in inset of Figure 1f. it would be good to also show the χ (divergence will be out of the range, but it will be still informative).

Because of the singularity when α approaches 1, the ratio does not exist and we decided not to add it to the inset.

3. It seems that k and K are used for the same thing. (449 and 450). Fixed.

Reviewer #1 (Remarks to the Author):

I'm satisfied with the revisions addressing the points that I raised in my previous reading of the manuscript. I also think the responses to the points raised by the other two reviewers are thorough and adequate. It's a great paper!

Reviewer #3 (Remarks to the Author):

I appreciate the authors' effort in improving statistics and methodology explanations. Notably, the scaling with a larger field of view looks much more convincing.

To the question of thresholding: I have read the derivations (that rely on the exact parabolic shape of the mean avalanche), and they are sound. However, the individual scaling exponents for sizes could be affected (as shown in Villegas et al.); you were not yet showing the sizes distribution (say, for the best k) in two different thresholding methods.

Thus, I would still ask you to put in the main text the avalanche-size definition, including only above the threshold part, because the current definition was shown to generate misleading results. I believe that it is essential for potential newcomers to the field they learn from your paper to use sound methodology.

Minor comment: the derivations and captions are written for Θ but in the panel Suppl 6. A. it is written as t . In the same figure, the inset "soft threshold," I guess, was planned to have the same line as in the "Hard threshold" (with straight lines down, below the S_{Θ}).

Synchronization: I got your point and can accept it, though for me, "synchronization" means something above the chance level for a given firing rate and duration (coarse-graining).

NCOMMS-22-02466B_Plenz

Reviewer #1 (Remarks to the Author):

I'm satisfied with the revisions addressing the points that I raised in my previous reading of the manuscript. I also think the responses to the points raised by the other two reviewers are thorough and adequate. It's a great paper!

Response: We thank the reviewer for these encouraging comments.

Reviewer #3 (Remarks to the Author):

I appreciate the authors' effort in improving statistics and methodology explanations. Notably, the scaling with a larger field of view looks much more convincing.

To the question of thresholding: I have read the derivations (that rely on the exact parabolic shape of the mean avalanche), and they are sound. However, the individual scaling exponents for sizes could be affected (as shown in Villegas et al.); you were not yet showing the sizes distribution (say, for the best k) in two different thresholding methods.

Thus, I would still ask you to put in the main text the avalanche-size definition, including only above the threshold part, because the current definition was shown to generate misleading results. I believe that it is essential for potential newcomers to the field they learn from your paper to use sound methodology.

Response: We now distinguish between the two forms of thresholding in main Figure 1e, where we first introduce our approach. A 'hard' threshold, which includes the area below the threshold, and a 'soft' threshold which removes it. This is now explicitly stated in the figure legend.

As requested, we have also added size distributions for soft and hard thresholding at the corresponding k at which $\chi_{sh} = 2$ (inset to Suppl. Fig. 6b). Distributions were calculated over all animals and experiments in the jRGECO1a group. We state in the corresponding legend:

Inset: Similarity in size distributions for the two thresholding approaches, obtained at the k for which $\chi_{sh} = 2$. The shallower slope for hard thresholding is expected given the higher k value compared to soft-thresholding (see also Fig. 1f), in addition to slope differences introduced by the thresholding method itself (Villegas et al., 2019; see main text).

We thank this reviewer again for allowing more space to establish the relationship in our data regarding soft and hard threshold. Indeed, we believe this is an important contribution to the field and deserves a follow up detailed analysis between thresholding, coarse-graining, and critical exponents, which though is beyond the scope of the current work.

Minor comment: the derivations and captions are written for Θ but in the panel Suppl 6. A. it is written as t . In the same figure, the inset “soft threshold,” I guess, was planned to have the same line as in the “Hard threshold” (with straight lines down, below the S_{Θ}).

Response: This is now corrected. Thank you for pointing this out.

Synchronization: I got your point and can accept it, though for me, “synchronization” means something above the chance level for a given firing rate and duration (coarse-graining).

Response: We understand this definition, though, an estimate of the firing rate itself is quite challenging without further assumptions. We therefore decided to identify the significance of these transients using controlled shuffling or addition of random spikes and demonstrate the degeneration of the scaling when pair-wise correlations are affected.